



# Seasonal variations of the backscattering coefficient measured by radar altimeters over the Antarctica Ice Sheet

Fifi I. ADODO[1, 2], Frédérique REMY[1], Ghislain PICARD[2]

[1]Laboratoire d'Etudes en Géophysique et Oceanographie Spatiale (LEGOS), Centre National de la Recherche Scientifique (CNRS), Toulouse, 31400, France
[2]Institut des Géosciences de l'Environnement (IGE), Grenoble, 38402, St Martin d'Heres Cedex, France

*Correspondence to* : Fifi I. ADODO (fifi.adodo@legos.obs-mip.fr)

**Abstract.** Spaceborne radar altimeter is a valuable tool for observing the Antarctica Ice Sheet. The radar wave penetration into the snow provides information both on the surface and the subsurface of the snowpack due to its dependence on the snow properties. However this penetration also induces a negative bias on the estimated surface elevation. Empirical corrections of this space and time-varying bias are usually based on the backscattering coefficient variability. We investigate the spatial and seasonal variations of the backscattering coefficient at the S (3.2 GHz), Ku (13.6 GHz) and Ka (37 GHz) bands. We identified two clearly marked zones over the continent, one with the maximum of Ku band backscattering coefficient in the winter and another with the maximum in the summer. To explain this, we performed a sensitivity study of the backscattering coefficient at the S, Ku and Ka bands to surface snow density, snow temperature and snow grain size using an electromagnetic model. The results show that the seasonal cycle of the backscattering coefficient at the Ka band, is dominated by the volume echo and is mainly explained by snow temperature. In contrast, the cycle is dominated by the surface echo at the S band. At Ku band, which intermediate in terms of wavelength between S and Ka bands, the seasonal cycle is in the first zone dominated by the volume echo and by the surface echo in the second one. Such seasonal and spatial variations of the backscattering coefficient at different radar frequencies should be taken into account the for more precise estimation of the surface elevation changes.

## 1 Introduction

Radar altimeters are within the most widely used sensors for measuring the surface elevation of polar ice sheets (Remy et al., 1999; Allison et al., 2009). It is a valuable tool for monitoring and quantifying change in volume of the Antarctic Ice Sheet (AIS) (Zwally et al., 2005; Wingham et al., 2006; Flament and Rémy, 2012; Helm et al., 2014). However, altimetric observations are affected by several errors: errors due to atmospheric and ionospheric propagations, slope error, or error due to radar wave penetration into the cold and dry snow (Ridley and Partington, 1988) that can be more or less corrected (Remy et al., 2012; Nilsson et al., 2016). Among all these potential errors, the latter is the most critical and the most challenging problem to tackle (Remy et al., 2012), as the distance observed between the satellite and the surface of the target is overestimated leading to a negative bias in the surface elevation estimation. For instance, Nilsson et al. (2015) have found a surface elevation bias of 0.5 to 1 m over the Greenland Ice Sheet. The temporal variation of this error is critical for scientific interpretation of ice sheet volume changes (Remy et al., 2012).

The signal recorded by radar altimeters, namely the waveform, is processed by an algorithm called "retracker" to determine several characteristics such as the range, the backscattering coefficient, the leading edge width and the trailing edge slope of the waveform. Various methods of waveform retracking exist, yet none adequately correct the effect of radar penetration (Arthern et al., 2001; Brenner et al., 2007). To reduce the temporal penetration variation error on the estimated surface elevation changes, Zwally et al. (2005) use an empirical linear relationship between the surface elevation and the backscattering coefficient time series at crossover points of the satellite tracks. Flament and Rémy (2012) use a non-linear relationship between time series of the surface elevation and the whole waveform parameters: the range, the backscattering





coefficient, the leading edge width and the trailing edge slope (computed with the ICE-2 retracker (Legresy et al., 2005)) on the along-tracks of the satellite. Both approaches are based on changes in the backscattering coefficient, which varies with time, reflecting changes of snowpack properties (Legresy et al., 2005; Lacroix et al., 2007). A more precise understanding of the annual and interannual variations of the backscattering coefficient is a prerequisite for improving the estimation accuracy of surface elevation trend over the AIS. In addition to measuring the surface elevation, the radar wave when penetrating the

snowpack provides information on the snow properties. Indeed, the backscattering coefficient is a combination of two components, the "surface echo" and the "volume echo" (Brown, 1977; Remy et al., 2012). The former mainly depends on surface roughness and density of near-surface snow while the latter mainly depends on snow temperature, grain size and snowpack stratification (Remy and Parouty, 2009; Li and Zwally, 2011) over a certain depth that mainly depends on the radar frequency (e.g. less than one meter at Ka band and less than ten meters at Ku band (Remy et al., 2015)).

The ENVIronment SATellite (ENVISAT) carries two radar altimeter sensors (RA-2) that operate at 13.6 GHz (Ku band) and 3.2 GHz (S band). The S band was originally intended for ionospheric corrections while the Ku band provides more accurate surface elevation due to the lower penetration depth. Comparison of the altimetric waveform characteristics between the Ku and S bands revealed different seasonal variations over the AIS (Lacroix et al., 2008b). The dual-frequency information can therefore be useful for retrieving information on snowpack properties. The launch in 2013 of the radar altimeter

SARAL/Altika that operates at the Ka band (37 GHz) and has the same 35-day phased orbit as ENVISAT allowed comparisons with much higher frequencies for the first time. Temporal variations of the estimated surface elevation with respect to the backscattering coefficient are 6 times lower at the Ka band than that of the Ku band, which implies that the volume echo at the Ka band comes from the near subsurface (<1 m) and is mostly controlled by ice grain size and temperature (Remy et al., 2015).

The radar wave penetration provides information on the snow properties, but it complicates the interpretation of the backscattering coefficient because more snow parameters are involved in the variation of the latter. To clarify the impacts of snow parameters on the backscattering coefficient, this paper investigates the spatial and seasonal variations of the radar backscattering coefficient at the S, Ku and Ka bands. To this end, electromagnetic models are used to assess the backscattering coefficient sensitivity to snow properties at the three frequencies. The aim of this paper is to determine snow

parameters, which dominate the seasonal cycle of the backscattering coefficient of each radar frequency, susceptible to affect empirical corrections applied to the surface elevation. This study is structured as follows: Section 2 presents the data, the calculation of the seasonal amplitude and date of maximum backscattering coefficient and depicts the radar altimeter electromagnetic models used to assess the seasonal variation of the surface and volume echoes at the S, Ku and Ka bands. Section 3 presents the spatial variations of the seasonal amplitude and date of maximum backscattering coefficient at the

three frequencies and the results of the sensitivity test of the volume echo with respect to snow density, snow temperature and snow grain size. Section 4 discusses the spatial distribution of the observed seasonal variability in the backscattering coefficient over the AIS.

## 2 Data and Methods

### 2.1 Altimetric Observations

Radar altimeter data were acquired by ENVISAT launched on March, 2002 by the European Space Agency (ESA). Acquisitions are simultaneous at the S and Ku bands, every 330 m along track on a 35-day repeat cycle orbit from September 2002 to October 2010 (the end of its repeat cycle orbit). The S band sensor failed after 5 years of measurements. The footprint has around 5 km radius and no data were acquired above 81.5° S due to ENVISAT's latitudinal orbit limit. Radar altimeter measures the power level and time delay of the radar echoes reflected by the snowpack - the so-called altimeter

echo or waveform - at a vertical sampling resolutions of 94 cm and 47 cm at the S and Ku bands, respectively.





To ensure post-ENVISAT mission and to complement the Ocean Surface Topography Mission (OSTM)/Jason (Steunou et al., 2015), the Satellite for ARgos and ALtiKa (SARAL)/AltiKa was launched on 25 February, 2013, by a joint CNES-ISRO (Centre National d'Etudes Spatiales - Indian Space Research Organisation) mission, on the same 35-day repeat cycle orbit as ENVISAT. On March, 2016, SARAL/AltiKa orbit was shifted onto a new orbit. Unlike classical Ku band radar altimeter,

the SARAL/AltiKa altimeter operates at the Ka band (37 GHz) and has a vertical sampling resolution of 30 cm. The ICE-2 retracking process was applied to the Ka band waveforms thus allowing estimation of the range, the backscattering coefficient ($\sigma^0$), the leading edge width and the trailing edge slope as for ENVISAT. The frequency ratios between the Ka and Ku bands, and the Ka and S bands are 2.7 and 11.6, respectively, which results in different sensitivity to the surface and the subsurface characteristics.

We processed 84 cycles of the backscattering coefficient from October 2002 until September 2010 for the Ku band and 55 cycles from October 2002 until December 2007 at the S band. Moreover, we consider 3 years of AltiKa altimeter data from March 2013 to March 2016, i.e. a total of 32 cycles of the backscattering coefficient over the whole Antarctic continent.

**2.2 Amplitude and date of maximum backscattering coefficient in the seasonal cycle**

The amplitude and the date at which the backscattering coefficient ($\sigma^0$) is at its maximum within a seasonal cycle were

calculated at the S, Ku and Ka bands for the entire Antarctic continent. Figure 1 shows an example of the temporal evolution of $\sigma^0$ at a location (69.46° S, 134.28° E) for the three bands. The time series of $\sigma^0$ exhibit a clear and well-marked cycle with a 1-year period (called seasonal cycle hereafter). The amplitude and the phase of the seasonal cycle $\sigma^0$ of were computed by fitting the time series of the observations with the following function Eq. (1):

$$\sigma_i^0(t) = \alpha_i \sin\left(2\pi\frac{t}{T}\right) + \beta_i \cos\left(2\pi\frac{t}{T}\right) + C_i, \qquad (1)$$

with $\quad A_i = \sqrt{\alpha_i^2 + \beta_i^2} \quad$ and $\quad \Phi_i = \arctan\left(\beta_i/\alpha_i\right);$

where $A_i$ and $\Phi_i$ are the amplitude and the phase of the seasonal cycle of $\sigma^0{}_i$, respectively, deduced from $\alpha_i$ and $\beta_i$, T = 365 days, $t$ ranges from 0 to 5 years for the S band, from 0 to 8 years for the Ku band and from 0 to 3 years for the Ka band with steps of 35 days and $i$ is the index of each along track data. Thus, we have a system of respectively 55, 84 and 32 equations for the S, Ku and Ka bands and three unknown parameters $\alpha_i$, $\beta_i$ and $C_i$, leading to a robust inversion. The fit was done with

the Ordinary Least Squares (OLS) method. Data were then gridded with a cell size of 5 km over the AIS.

**2.3 Backscattering coefficient modeling**

To identify the snowpack properties that are responsible for the seasonal cycle of $\sigma^0$, we investigated its sensitivity to the snowpack surface and subsurface properties using an altimetric echo model on snow. This model account for the surface echo and the volume echo. The surface echo results from the interactions of the radar wave with the snow surface (air-snow

interactions) while the volume echo results from the interactions of the radar wave with the scatterers within the snowpack (snow-snow interactions). The physics involved in both surface and volume echoes have been previously detailed for the AIS by Lacroix et al. (2008b) and Remy et al. (2015) and are depicted in Sect. 2.3.1 and 2.3.2.

**2.3.1 Surface echo modeling**

Snow surfaces may be modeled as randomly rough surfaces because most naturally occuring surfaces are irregular. The

surface scattering coefficient from rough surface is thus controlled by the effective dielectric constant of the medium and the surface roughness characteristics (Ulaby et al., 1982; Fung et al., 1994). The snow effective dielectric constant is a function of snow density and ice dielectric constant, while the roughness is usually prescribed by two statistical geometric parameters:



the surface correlation length (l) and the standard deviation of the surface height ($\sigma_h$) (Ulaby et al., 1982). In the case of large standard deviations of the surface height ($\sigma_h$) compared to the radar wavelength, the backscattering coefficient from

rough surface $\sigma_{sur}^0$ can be estimated assuming the roughness has a Gaussian auto-correlation function as follows (Ulaby et al., 1982):

$$\sigma_{sur}^0 = \frac{|R(0)|^2}{2S^2} ,$$ (2)

where $R(0)$ is the Fresnel reflection coefficient at normal incident and $S = l/\sigma_h$ the root mean square (RMS) of the surface slope at the radar wavelength scale. Equation (2) is almost independent of the radar wave frequency and $\sigma_{sur}^0$ increases with

increasing surface snow density and decreasing surface slope RMS. When the surface snow density increases from 300 to 400 kg m$^{-3}$, $|R(0)|^2$ increases from 1.27 10$^{-2}$ to 2.10 10$^{-2}$, resulting in variations of the surface echo from -1.97 dB to 0.21 dB for a given surface with a slope of 0.1. Surface snow density variations from 300 to 400 kg m$^{-3}$ induce a variation of ±2.17 dB in the surface echo.

**2.3.2 Volume echo modeling**

The volume echo is mainly controlled by the scattering coefficient ($K_s$), depending on the size of the scatterers and the radar frequency. The power extinction in the snowpack is the sum of the scattering coefficient ($K_s$) and the absorption ($K_{ab}$) coefficient. The latter depends on snow temperature and radar frequency. In the following, the scatterers are assumed to be spherical. The scattering coefficient ($K_s$) and the absorption coefficient ($K_{ab}$) are given by Mätzler, (1998):

$$K_s = \frac{3}{32} p_c^3 k_0^4 v (1-v)(\epsilon_i^{'} - 1)^2 K_d^2 ,$$ (3)

$$K_{ab} = k_0 v \epsilon_i^{''} K_d^2 ,$$ (4)

where $k_0 = 2\pi/\lambda$ is the wave number and $\lambda$ the wavelength, $v$ is the fractional volume of the scatterers, $\epsilon_i^{'}$ and $\epsilon_i^{''}$ are the real and imaginary parts of the effective dielectric constant of pure ice, $p_c = (4r_g)/3$ (Mätzler, 1998) is the correlation length (used here as the effective size parameter) with $r_g$ the scatterers radius and $K_d^2 = |2\epsilon^{'} + 1|^2 / |2\epsilon^{'} + \epsilon_i^{'}|^2$ with $\epsilon^{'}$ the real part of the effective dielectric constant of snow (Tiuri et al., 1984).

For snow grain radius increasing from 0.3 to 0.5 mm, $K_s$ increases from 1.05 to 4.85 m$^{-1}$ at the Ka band, from 0.02 to 0.08 m$^{-1}$ at the Ku band, and from 0.58 10$^{-4}$ to 2.7 10$^{-4}$ m$^{-1}$ at the S band. As snow temperature varies from 220 to 250 K, $K_{ab}$ increases from 0.194 to 0.287 m$^{-1}$ at the Ka band, from 0.026 to 0.039 m$^{-1}$ at the Ku band, and from 0.002 to 0.003 m$^{-1}$ at the S band. The extinction coefficient at the Ka band is dominated by the scattering coefficient. In contrast, the losses by absorption dominate the extinction at the S band while at the Ku band, both coefficients are of the same order of magnitude.

Volume scattering mainly affects the Ka and Ku bands. Finally, the losses by absorption increase with snow temperature while the scattering coefficient is mainly driven by snow grain size. Both the losses by absorption and scattering coefficient increase with increasing radar frequency.

- *Snow property profiles*

For all the simulations, we considered the same vertical density profile as Lacroix et al. (2008b) with a variation only in the

top first 10 m given by :

$$\rho(z) = \rho_0 + p z + c_2 z^2 + c_3 z^3 .$$ (5)



where $c_2$ and $c_3$ are constant values taken from the Talos Dome density profile, $-1.35\ 10^{-4}$ and $5.86\ 10^{-7}$, respectively (Frezzotti et al., 2004), $\rho_0$ is the mean surface density and p $= 1.40\ 10^{-2}$ is calculated as a function of $\rho$ so that the density at the depth below the surface $z = 10$ m is the density measured at the Talos Dome (72.78° S, 159.06° E). Snow temperature is computed using the solution of the thermal diffusion equation (e.g Bingham and Drinkwater, 2000; Surdyk, 2002), assuming a sinusoidal seasonal surface temperature and constant snow thermal diffusivity $\kappa$. The temperature at depth z is of the form:

$$T(z,t) = A_m e^{\left(\frac{-z}{l}\right)} \cos\left(\omega t - \frac{z}{l}\right) + T_m,\qquad\qquad (6)$$

where $Am$ and $Tm$ are the seasonal amplitude and mean temperatures, respectively, $\omega$ is the angular frequency, $t$ is the time, $z$ is the depth and $l = \sqrt{2\kappa/\omega}$. $\kappa$ is the ratio of the thermal conductivity ($\kappa_d$) to the heat capacity and the snow density ($\rho$).

We used the quadratic relationship of the thermal conductivity derived by Sturm et al. (1997): $\kappa_d = 0.138 - 1.01\rho + 3.233\rho^2$. In the computing of $\kappa_d$ and $\kappa$, snow density, $\rho$, is assumed equals to an average of the density profile of Eq. (5) (Bingham and Drinkwater, 2000). The temperature wave propagating in the snowpack has decreasing amplitude with respect to depth. The snow grain growth rate is mainly dependent on snow temperature (Brucker et al., 2010) and the snow grain profile with depth (Bingham and Drinkwater, 2000) is expressed  by:

$$r_g(z)^2 = r_0^2 + k_g z/\pi D,\qquad\qquad (7)$$

where $k_g = 0.00042$ mm$^2$ yr$^{-1}$ is the typical snow grain growth rate, $D$ is the mean annual snow accumulation (mm yr$^{-1}$), $z$ is the depth and $r_0$ is the spherical scatterer mean radius at the surface. Tests of variation of D show no significant effect on the volume echo trend, and we therefore set D to 50 mm yr$^{-1}$ (Bingham and Drinkwater, 2000).

## 3 Results

### 3.1 Spatial patterns of the amplitude and date of maximum backscattering coefficient

The spatial distribution and the histogram of the seasonal date of maximum σ⁰ at the S, Ku and Ka bands are shown in Fig. 2 and Fig. 3, respectively. Among the three bands, the Ku band  presents the most contrasted geographical patterns. In the zone that appears in yellow, the seasonal cycle of σ⁰ reached its maximum early in the year (summer peak zone, SP hereafter). This zone covers the Eastern central part of the AIS which encompasses the domes and high altitudes regions (~ > 3000 m

asl). It extends from Wilkes Land to Dronning Maud Land (DML) and is characterized by a decrease in σ⁰ from late autumn to early spring followed by an increase at the end of the summer. The zone appearing in blue (hereafter winter peak zone, WP), encompasses the lower regions (< 3000 m asl) including coastal steeply-sloped regions. It is characterized by an increase in σ⁰ from late autumn to early spring. In contrast to the Ku band, the seasonal cycles of σ⁰ over the AIS are generally, maximum in the summer at the S band whereas they are maximum in the winter at the Ka band. In Fig. 3, the Ku

band date of maximum σ⁰ histogram is clearly bimodal with peaks between Julian days 1 and 100 and between Julian days 175 and 275. In the following, these two periods are referred to as summer and winter, respectively. With these definitions, the WP and SP zones represent 42% and 45% of the AIS, respectively. The histogram of the date of maximum σ⁰ at the S and Ka bands are unimodal with a peak in summer for a lower frequency (WP : 11%, SP : 66%)  and a peak in winter for a higher frequency (WP : 50%, SP : 14%). The difference of the seasonal date of maximum σ⁰ between the Ku and Ka bands

(Fig. 4), over the AIS, shows a geographical pattern similar to that observed in Fig. 2b. Negative values indicate that σ⁰ is maximum at the Ku band before the Ka band while positive values indicate the opposite. Negative values account for about 36% of the AIS and coincide with the SP zone where σ⁰ is maximum in summer at the Ku band. Positive values, the zone appearing in blue, cover 48% of the AIS and are correlated to the WP zone. Hence, we note a positive lag of the date of



maximum $\sigma^0$ between the Ku and Ka bands only in the zone where $\sigma^0$ is maximum in the winter in both frequencies and a
negative lag in the other zones. The spatial distribution of the seasonal amplitude of $\sigma^0$ at the Ka band (Fig. 5c) shows an
obvious geographical pattern close to that of the seasonal date of maximum $\sigma^0$ at the Ku band. The Ka band seasonal
amplitude of $\sigma^0$ is the highest in the WP zone (1.02 ± 0.56 dB) and weakest in the SP zone (0.53 ± 0.41 dB) as shown in
Fig. 6. By contrast, the seasonal amplitude of $\sigma^0$ at the S band (Fig. 5a) appears anti-correlated with that at the Ka band,
exhibiting a large seasonal amplitude in the SP zone (0.79 ± 0.40 dB) and a weak amplitude in the WP zone (0.42 ± 0.28
dB). The seasonal amplitude of $\sigma^0$ in the SP zone is almost twice as large as that of the WP zone at the S band and the
inverse is true at the Ka band. The seasonal amplitude of $\sigma^0$ at the Ku band shows no evident regional patterns and is almost
of the same magnitude in both zones (Fig. 5b), except in the interior of Wilkes Land, Princess Elisabeth Land and the Ronne
Ice Shelf, which showed the maximum amplitudes.

### 3.2 Temporal variations of the surface elevation with respect to the backscattering coefficient

Figure 7 shows the spatial distribution of temporal variations of the estimated surface elevation with respect to $\sigma^0$ at the Ku
band, hereafter denoted dhd$\sigma^0$. Negative values of dhd$\sigma^0$ indicate that surface elevation decreases when $\sigma^0$ increases,
implying that temporal variations of $\sigma^0$ are due to changes in the deep snowpack properties, i.e. in the volume echo. On the
contrary, positive values of dhd$\sigma^0$ indicate that the surface elevation increases with $\sigma^0$. In this case, the temporal variation of
$\sigma^0$ are due to changes in the surface echo. The map in Fig. 7 shows that near-zeros and negative values of dhd$\sigma^0$ (in blue) are
found in the WP zone. This means that the WP zone undergoes large variations of volume echo.

### 3.3 Sensitivity test

Since there are few if any studies on the seasonal cycle of the snow surface roughness, it is poorly known. The sensitivity
study of the surface echo is thus limited by the lack of information on snow surface roughness, in particular over the AIS.
Consequently, we have focused on the modeling of the seasonal cycle of the volume echo. In this subsection, the sensitivity
test of the volume echo at the S, Ku and Ka bands to snow properties is explored considering three parameters snow
temperature, snow grain size and snow density in the analysis of the seasonal cycle of $\sigma^0$.
The model shows an increase in the volume echo with snow density at the three frequencies (Fig. 8a). Snow density controls
the thermal conductivity of the medium. Increasing surface snow density increases thermal diffusivity, which attenuates the
propagation of the temperature wave in the snowpack. Figure 8 (b and c) shows that the volume echo at the S band is not
sensitive to snow temperature and grain size variations, while the volume echo at the Ku and Ka bands is affected by both
parameters. Snow density, temperature and grain size impacts on volume echo are more significant at the Ka band than at the
Ku and S bands. The volume echo increases with the snow density at the three frequencies, and at the S band the volume
echo is less significant.

### 4 Discussion

The sensitivity of the volume echo to snow temperature shown in Fig. 8b implies that the volume echo is maximum in winter
at the Ku and Ka bands and constant at the S band. This sensitivity is explained by the fact that increasing snow temperature
increases absorption resulting in a decreases of the radar wave in the volume, thus limiting the volume echo. Also increasing
snow grain size increases the scattering coefficient, which increases the radar wave extinction in the snowpack, and
conversely decrease the radar wave penetration, therefore may affect the volume echo. Moreover, the positive lag observed
between the Ku and Ka bands in the WP zone in Fig. 4 can be explained by the difference of the radar wave penetration
depth between the Ku (~10 m) and Ka (>1 m) bands in the snowpack. This lag is related to the propagation of the



temperature wave from the surface to the subsurface of the snowpack. The volume echo variations is therefore predominantly driven by the seasonal variations of snow temperature in the WP zone.

Snow density is involved in both the surface and volume echoes. These echoes increase with increasing surface snow
density, thus similar seasonal cycle of $\sigma^0$ would be expected at any frequency if snow density were the main driver. This is in contradiction to the observations (Fig. 3). Therefore the seasonal cycle of $\sigma^0$ cannot be explained solely by snow density. Being insensitive to snow temperature and grain size (Fig. 8b,c), the seasonal cycle of $\sigma^0$ observed at the S band cannot be explained by the volume echo. This implies that snow surface properties (surface snow density and roughness) are the main factors driving the seasonal cycle of $\sigma^0$ at the S band.

The dry snow of inland Antarctica is heterogeneous medium consisting of a mixture of air and ice crystals similar to dry soil, i.e. a mixture of air and solid soil material. Fung (2010) explains that a soil surface acts like a surface at centimeter wavelength. But when the wavelength is shortened to less than a millimeter, the surface appears to the sensor as a dense collection of scatterers sitting above another surface or  simply as a volume-scattering medium because  the individual sand grains of the soil surface are being seen by the sensor. From the S to Ka band, the radar wavelength decreases by a factor 12
from 9.4 cm to 0.8 cm corresponding to a scale change from centimeter to millimeter. We assume that the snow surface and the soil surface behave in the same way. This means that the snow surface is sensed as a surface at the S band and as a volume-scattering medium at the Ka band. The latter is particularly true because snow grain size is comparable to the Ka band radar wavelength. In addition, the volume echo variation is greater at the Ka band than at the S band. Therefore, w e argue that the seasonal cycle of $\sigma^0$ observed at the Ka band is dominated by that of the volume echo. This explains that the
maximum $\sigma^0$ is observed in winter at the Ka over the AIS.

Several observations show that sastrugi (10 cm to 1 m height) are the main contributors to surface roughness (Kotlyakov, 1966; Inoue, 1989; Lacroix et al., 2007). Since the biggest features (hectometer to kilometer scales) change little over time, it is likely that the most influential roughness scale in the seasonal cycle of the surface echo is the sastrugi on the surface (Lacroix et al., 2008a). Despite the increase in surface and volume echoes with surface snow density, evidences from Fig. 3
suggests that the seasonal cycle of $\sigma^0$ cannot be explained by the seasonal cycle in surface snow density. Therefore, it is likely that the seasonal cycle of $\sigma^0$ observed at the S band, predominantly driven by the surface echo, stems from the seasonal cycle of snow surface roughness. However, in this study it is difficult to differentiate with certainty which one among the surface snow density or the snow surface roughness, dominates the seasonal cycle of the surface echo. (i) The snow surface roughness is poorly known and in particular its seasonal variability; (ii) surface snow properties evolve rapidly
with the wind and (iii) the surface snow roughness and density are interdependent and linked because the denser the snow surface, the larger the effect of surface roughness due to the increase of the effective dielectric discontinuity (Fung, 1994).

Considering that $\sigma^0$ at the Ku band shows two opposing seasonal cycle patterns over the AIS and its wavelength is between that of the S and Ka bands, we suggest that $\sigma^0$ at the Ku band is dominated by the seasonal cycle of the surface echo, similar to the S band in the SP zone and by the seasonal cycle of the volume echo, similar to the Ka band in the WP zone. We
support this hypothesis with ancillary data and by modeling. By overlaying the Antarctica radarsat mosaic with the SP zone contours (Fig. 9), we find that the WP zone matches with regions of greatest heterogeneous backscatter of radarsat, where megadunes (Frezzotti et al., 2002) and wind-glazed surfaces (Scambos et al., 2012) have been observed. The seasonal cycle of $\sigma^0$ at the Ku band is maximum in the winter in heterogeneous radarsat backscatter regions while it is maximum in the summer in the other regions. In fact, areas of megadunes are characterized by slightly steeper regional slope and the presence
of highly persistent katabatic winds (Frezzotti et al., 2002) and wind-glazed surfaces have been formed by persistent katabatic winds in areas of megadunes (Scambos et al., 2012). There exists therefore a relationship between the wind and the seasonal cycle of $\sigma^0$. To further investigate this point, we used ERA-Interim reanalysis wind speed data supplied by ECMWF (European Centre For Medium-Range Weather Forecasts) on the period corresponding to that of Ku band. We observe a high spatial coherence of the seasonal amplitude of the wind speed (Fig. 10a) patterns with the date of maximum $\sigma^0$ over the



seasonal cycle at the Ku band (Fig. 2b). Wind speed average ($8.2 \pm 1.6$ m s$^{-1}$) and seasonal amplitude ($1.7 \pm 0.4$ m s$^{-1}$) are higher in the WP zone than in the SP zone ($6.6 \pm 1.58$ m s$^{-1}$ and $1.0 \pm 0.3$ m s$^{-1}$, respectively).

The striking similarity in the spatial distribution of the seasonal amplitude of $\sigma^0$ at the Ka band (Fig. 5c) and the seasonal date of maximum $\sigma^0$ at the Ku band (Fig. 2b), which is itself correlated to the seasonal amplitude of the wind speed (Fig. 10a) suggests that the wind plays a significant role in the distribution of the seasonal amplitude of $\sigma^0$ at the Ka band. Although the wind effects on the snowpack are numerous and complex, we retained two for which we simulated the impacts on the volume echo (Fig. 8):

- a) Wind may smash snow grains so that the surface snow density increases with wind speed (Male, 1980); this leads to an increase in the amplitude of the volume echo at the three frequencies as shown in Fig. 8a. Surface snow density is a good candidate for explaining the spatial distribution of the seasonal amplitude of $\sigma^0$ at the Ka band because snow compaction can occur at different times of the year depending on the snow accumulation rate and the temperature gradient (Li and Zwally, 2002 and 2004).

- b) Increasing wind speed leads to an increase in blowed snow transport, that removes all or almost all the precipitated or wind deposited snow that may temporarily accumulate (Scambos et al., 2012; Lenaerts et al., 2012). This implies that there is no significant change in the surface mass balance over an annual cycle, i.e. near-zero net accumulation (Scambos et al., 2012), allowing snow surface to be almost constant. This corroborates our contention that the seasonal variation of $\sigma^0$ at the Ku band in the WP zone emanates exclusively from the volume echo of the lower layer (Fig. 7). Thus it is presumably that these variations are due to depth hoar formation during winter in the WP zone. Indeed, the wind speed is on average maximum between Julian days 170 and 230, when air temperature is colder than the snow temperature. By blowing on the snowpack, cold and persistent winds unusually accelerate the cooling of the surface snow temperature (Remy and Minster, 1991). This causes an important temperature gradient, which determines the rate of metamorphism of snow grains within the snowpack. This specific increase of the temperature gradient promotes depth hoar formation in winter, that creates coarse cup-shaped ice crystals (Scambos et al., 2012), acts as more effective volume-scatterers and hence increase the volume echo as predicted in Fig. 8c. For instance, Brucker et al. (2010) have found the highest grain size vertical gradient in the regions of the WP zone.

Finally, the combined effects of wind speed and temperature may explain the difference observed between the seasonal cycle of $\sigma^0$ at the Ka and Ku bands. Similarly, the spatial distribution of the seasonal amplitude of $\sigma^0$ at the Ka band is ascribed to the wind effects mentioned above on the snowpack.

## 5 Conclusion

The radar altimeter remaining on the same tracks with the same 35-day revisit time allowed to carry out this spatial and temporal comparative study of the seasonal amplitude and date of maximum $\sigma^0$ at the S, Ku and Ka bands. We used 8-year long time series of $\sigma^0$ for the Ku band,  5-year long time series of $\sigma^0$ for the S band and  3-year long time series of $\sigma^0$ for the Ka band all covering 2002 to 2010 for ENVISAT sensors and 2013 to 2016 for the SARAL/Altika sensor. The backscattering coefficient shows seasonal variations with varying amplitude and phase over the AIS and with a marked dependence to radar frequency. In general, it is maximum in winter at the Ka band, and maximum in summer at the S band. At the Ku band, both behaviors are found on the AIS, maximum in winter in the so-called WP zone and maximum in summer in the SP zone.

We investigated the snow properties that dominate volume echo seasonal changes with electromagnetic models of the backscattering coefficient. As a result, we showed that variations in the snow properties, such as temperature and grain size, cannot explain the seasonal cycle of $\sigma^0$ observed at the S band because of its small sensitivity to those properties. In contrast, the temperature cycle may well explain the seasonal cycle of $\sigma^0$ at the Ka band. We explain that the contrasted seasonal cycle





of $\sigma^0$ observed at the Ku band, which is between the S and Ka bands, is due to its high sensitivity to the volume echo in the WP zone and to the surface echo in the SP zone. The geographical pattern of the WP and SP zones is related to the seasonal amplitude of the wind speed and may therefore be a consequence of the presence or not of wind-glazed areas induced by strong persistent winds.

These results should be considered to mitigate radar induced penetration error on the estimated elevation variations and improve the accuracy of the Antarctic surface mass balance for three main reasons: (i) the choice of the radar wavelength is very important to reduce the sensitivity to changing snowpack properties; (ii) altimetric waveforms will be better interpreted according to the frequency and the location; and (iii) at the Ku band, particular attention should be paid to the WP zone which undergoes large variations of snow properties. Multi-frequency sensors are the key to improve the understanding of

the physics of radar altimeter measurements over the AIS. An important limitation of this study is the lack of information on the seasonal variability of the snow surface roughness in Antarctica, which will be the topic of future work.

### Acknowledgements

This work is a contribution to the ASUMA (improving the Accuracy of the SUrface Mass balance of Antarctica) project funded by the Agence Nationale de la Recherche, contract ANR-14-CE01-0001-01. ENVISAT and AltiKa data were

provided by the Center for Topographic studies of the Oceans and Hydrosphere (CTOH) at LEGOS and are available at http://ctoh.legos.obs-mip.fr/. The authors would like to thank Etienne Berthier and Jessica Klar from LEGOS for their helpful comments and suggestions.

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



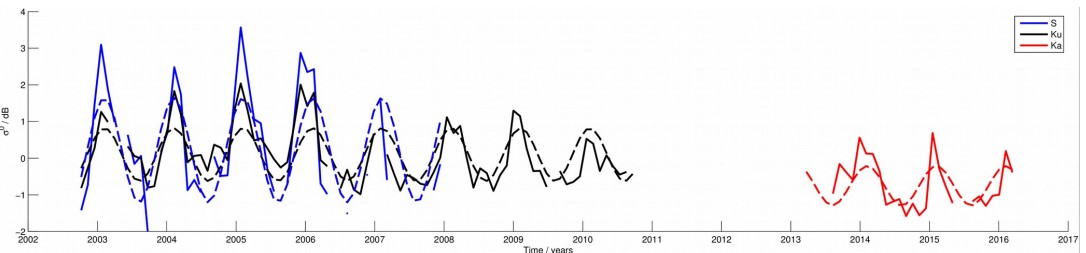

**Figure 1: Time series of the backscattering coefficient at the S (blue), Ku (black) and Ka (red) bands at a location (69.468°S, 134.28°E). Dashed lines represent the fits. The observations show seasonal cycle with a 1 year period.**

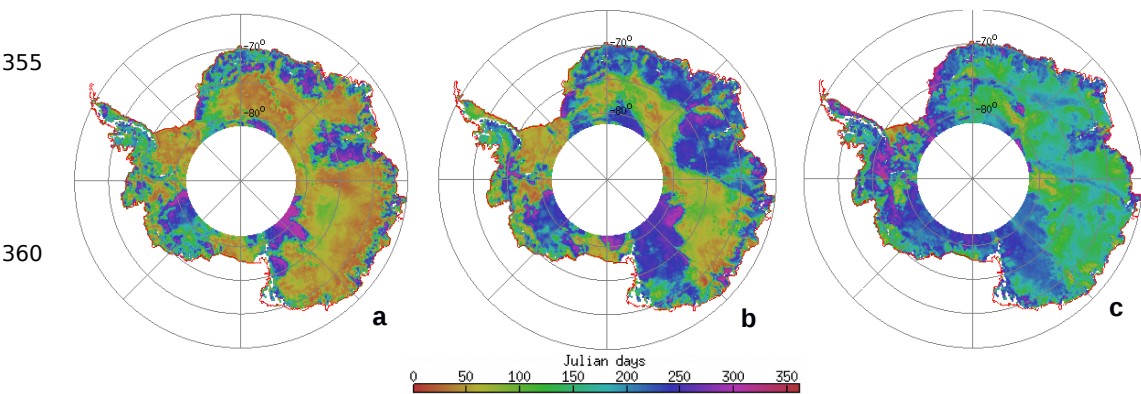

**Figure 2: Spatial distribution of the seasonal date of maximum backscattering the coefficient at S (a), Ku (b) and Ka (c) bands. The colorbar is cyclic and in Julian days.**

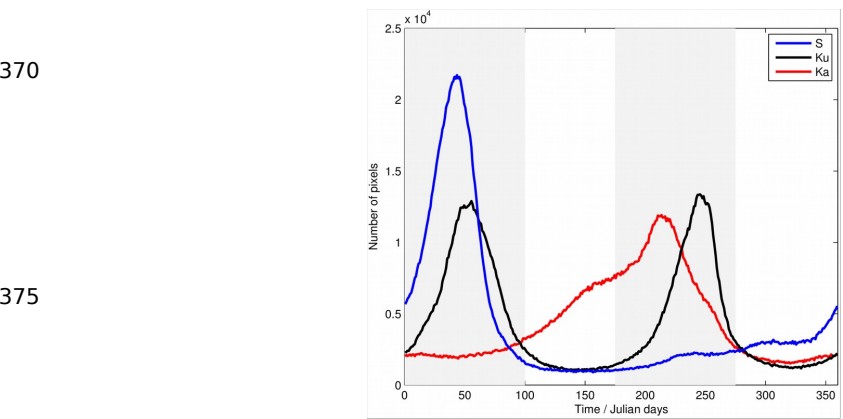

**Figure 3: Histogram of the seasonal date maximum backscattering coefficient at the S (blue), Ku (black) and Ka (red) bands. The gray bars represent periods referred to as summer and winter.**






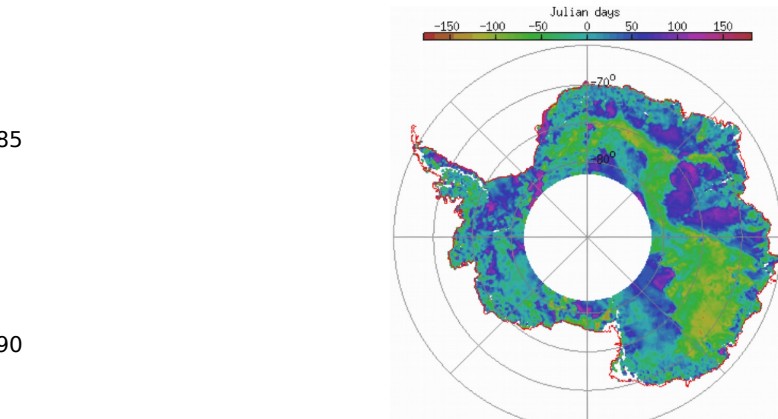


**Figure 4: Difference of the seasonal date maximum backscattering coefficient between the Ku and Ka bands. The colorbar is cyclic and in Julian days.**



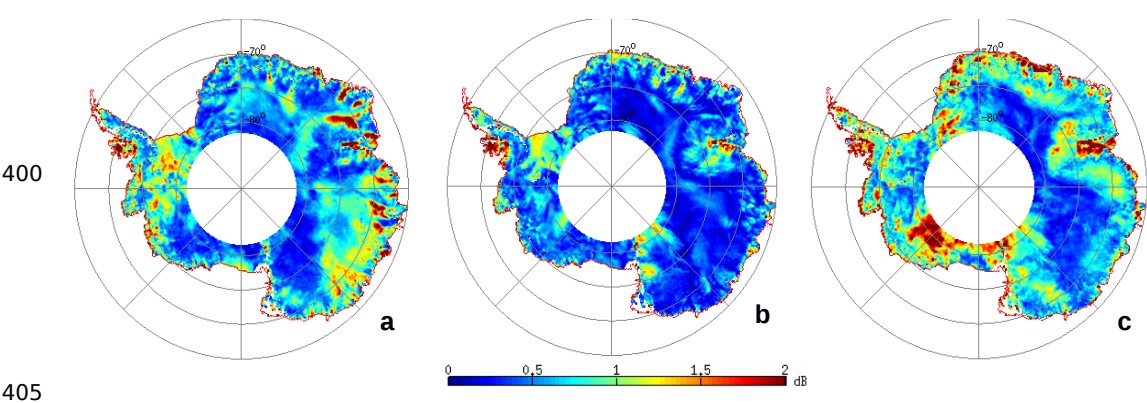


**Figure 5: Spatial distribution of the seasonal amplitude of the backscattering coefficient at the S (a), Ku (b) and Ka (c) bands. Values are expressed in dB.**



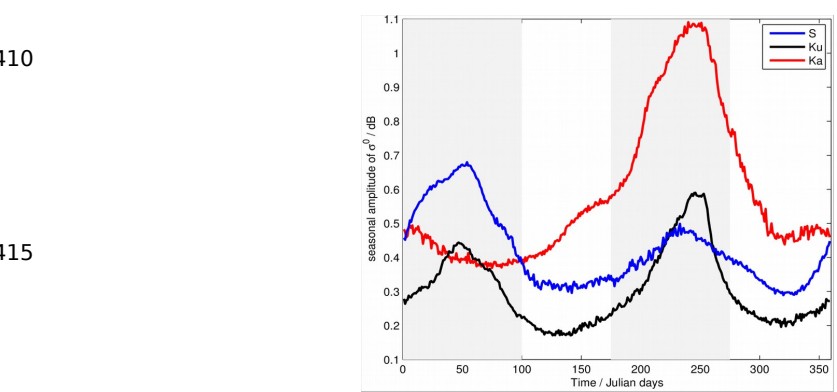

**Figure 6: Mean seasonal amplitude with respect to the date of maximum backscattering coefficient at the S (blue), Ku (black) and Ka (red) bands. The gray bars represent periods referred to as summer and winter.**




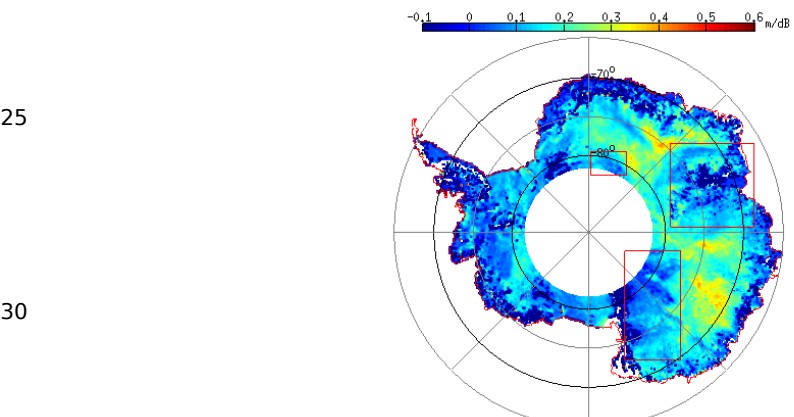

**Figure 7: Temporal variations of the surface elevation with respect to the backscattering coefficient at the Ku band (dhdσ⁰). Red boxes show regions where this parameter is negative or close to zero. Values are expressed in dB.**

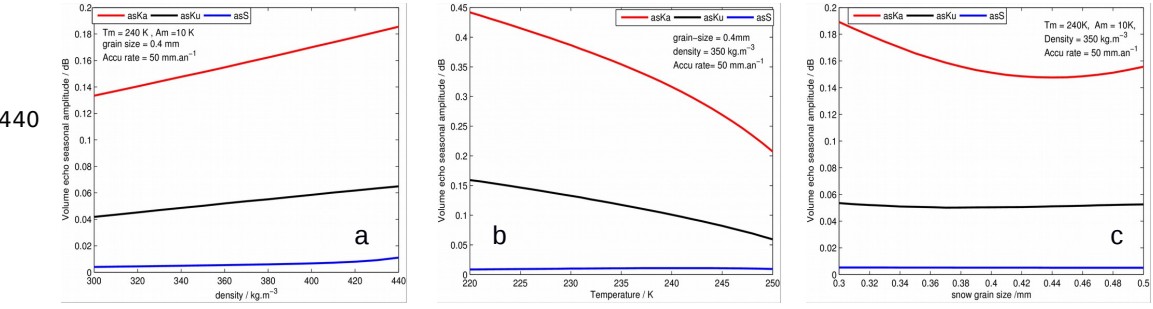

**Figure 8: Sensitivity study of the volume echo with the surface snow density (a), snow temperature (b) and snow grain size (c) at the S (blue), Ku (black) and Ka (red) bands.**

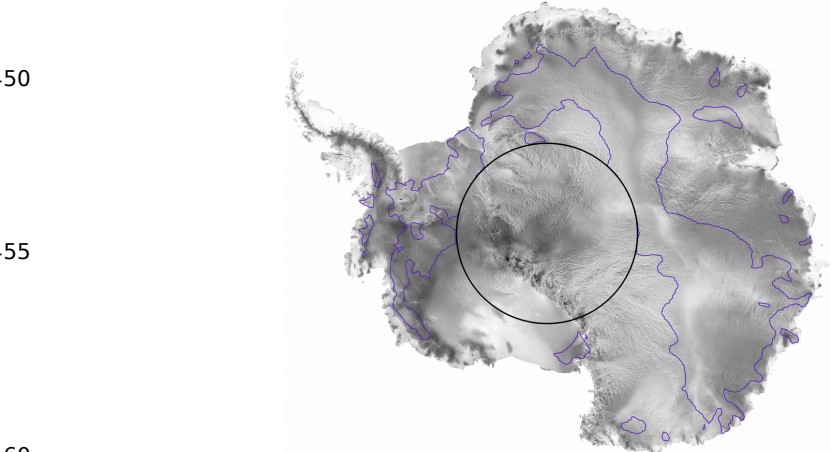

**Figure 9 : Distribution of the date of maximum backscattering coefficient at the Ku band superimposed on the Radarsat mosaic (RAMP). Contours in blue show the borders between the WP and the SP zones over the Antarctica Ice Sheet. SP zone, regions where the backscattering coefficient is maximum in summer, is inside the contours and the WP zone where the backscattering coefficient is maximum in winter is situated outside. No data are available beyond 81.5° S (black circle).**





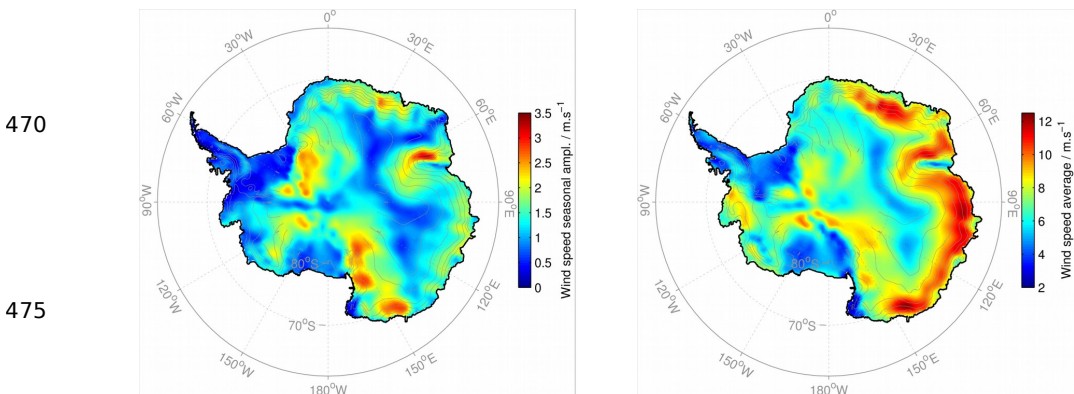



**Figure 10 : Seasonal amplitude (left) and average (right) of wind speed. Data are extracted from ERA-Interim reanalysis provided by ECMWF, on the period corresponding to that of ENVISAT lifetime and are gridded at $25 \times 25$ km² data before computing the average and amplitude. Thin gray contours are 500 m asl elevation intervals.**
