# Peer review of "Seasonal variations of the backscattering coefficient measured by radar altimeters over the Antarctic Ice Sheet"

_The Cryosphere, 2017_

## Referee Comment (RC1) · Anonymous Referee #1 · 29 Dec 2017

In this study, Adodo et al. observe seasonal variations in radar backscatter over the Antarctic Ice Sheet using three different radar frequencies. The authors define regions over Antarctica where backscattered power is found to peak in the summer for the S band, winter for the Ka band, and in both summer and winter for the intermediate Ku band. The authors perform a sensitivity study to help understand the effects of surface snow density, snow temperature, and snow grain size on backscattered power from each radar band. This study, in particular the delineation of these summer and winter 'peak zones', as referred to by the authors, represents a worthwhile addition to the literature. However, in my opinion some of the reasoning the authors provide in the discussion section to relate these seasonal variations to physical processes is lacking

in places, and I would appreciate if they could respond to the following comments.

Specific Comments:

Section 1

P1 L8: I would suggest rephrasing to "radar wave interaction with the snow..." instead of "radar wave penetration...".

P1 L27: I feel the phrase "More or less corrected" is quite vague here: corrections for atmosphere/ionosphere and slope errors are well established in the literature and minimise these errors with good accuracy – radar wave penetration is the main outstanding problem listed.

P1 L30: I have some concerns with the 2012 Greenland melt event being used as an example here. This is a positive elevation bias (not negative as the authors discuss in the preceding sentence) caused by a resetting of the radar scattering horizon due to an anomalous surface melt event. This is a process without equivalent in Antarctica, and one that has also been corrected for in the literature when measuring surface elevation change with radar altimetry (Nilsson et al., 2016, McMillan et al., 2016). In my opinion the authors should clarify this here, or include more examples on the effects of radar penetration on time series of elevation in Antarctica from radar altimetry in order to better establish the problem they are addressing.

P1 L41: The authors fail to mention the work of Davis and Ferguson here (Davis and Ferguson, 2004), however I feel this is a significant contribution to the literature which the authors should include.

P2 L46: The authors should cite Ridley and Partington 1988 here.

P2 L50: I think it would be helpful for the reader to know the wavelength of each radar band in addition to the frequency, either here on in Section 2.1.

P1 L55: The manuscript states later on that the orbit of AltiKa has recently been shifted,

so I would suggest amending this sentence to reflect this.

Section 2

P2 L79: The authors describe the radar waveform, but the concept has already been introduced in the previous section. I would suggest formally defining the waveform where it is first mentioned as opposed to here, as it is a key concept needed for the paper.

P3 L87: The authors should rephrase this sentence to make it more clear that the ICE-2 retracker is used to obtain backscattering coefficients for the Ku and S bands.

Do the authors consider ascending and descending tracks separately? A previous study has shown radar backscatter has an anisotropic dependence resulting from the interaction between the radar polarization direction and wind induced features of the firn (Armitage et al., 2013).

Figure 1: Can the authors please include a map to indicate where this location is in relation to the continent, and along the orbit tracks of the 3 bands.

P3 L105: Do the authors place any controls on poorly constrained fits due to e.g. poor match between observed and modelled seasonal peaks?

In addition, can the authors please provide more information on how the amplitude and phase are gridded. Do they use the mean? If so, are there grid cells which have a high variance? How many coefficients, on average, are binned into a 5 km grid cell for each radar band? What data coverage does this provide in more challenging regions such as the margins and the Peninsula?

P4 L151: Can the authors please comment on the validity of applying this firn density profile obtained at one location to the rest of the Antarctic ice sheet – how sensitive are the results to this assumption?

Section 3

P5 L168: Which day/month of the year do these peaks correspond to? More information can be provided to the reader here.

Figure 2: It would be useful to plot elevation contours (or an inset elevation map) if elevation is used to delineate backscatter patterns in the text. It would also be helpful for the authors to indicate the locations of regions they refer to in the text (e.g. Wilkes Land, Dronning Maud Land).

Would it also be possible for the authors to mark out the summer and winter peak zones they define in the text? Finally, I would also suggest using a different colour scale, the differences between pale yellow-green-blue are quite hard to make out.

P5 L177: Do these percentages refer to the observed area, or the entire Antarctic ice sheet? This applies to any percentage stated in this way.

P5 L178: Do the percentages in brackets also refer to the area of these summer and winter zones for each band? As written it is not clear to me, I would suggest rephrasing this.

Figure 4: I would suggest using a different colour scale which is preferably divergent to make the figure clearer.

P6 L187: How are the uncertainties in backscatter coefficient derived here? They appear to be quite large to me.

P6 L195: Can the authors please expand on how they are deriving surface elevation, is it also from the ICE-2 retracker, and binned at the same 5 km grid used for the backscatter coefficients? Have these elevations been corrected for atmosphere/slope? In addition, how are the values of $dhd\sigma$ derived?

Figure 7: The units in the caption state dB not m/dB.

Section 4

P6 L217: "...resulting in a decreases of the radar wave in the volume...". Are the

authors referring to a decrease of backscattered power? I suggest the use of more precise language in instances like this. Also decreases should be decrease.

P6 L220: It would be helpful to show this WP zone on Figure 4 to make clear to the reader.

P7 L222: Do the authors have any evidence to back up this assertion of volume echo variations being driven by temperature? I agree this is a reasonable conclusion to propose, however the authors do not offer enough evidence to convince that this is indeed the case. I would suggest that the sentence is reworded to make the authors argument clearer.

P7 L230-240: I am not sure I agree with the soil analogy – in my opinion it doesn't offer any clarity to the reader and isn't needed. Can the authors please expand on what they mean when they state the snow surface is sensed "as a volume scattering medium at the Ka band" – in reality there will always be a surface component of the radar echo controlled by incidence angle and topography on the footprint scale.

P7 L244: Do the authors mean to reference Fig. 8 here and not Fig. 3?

P7 L246: Do the authors have any evidence for a seasonal cycle of snow surface roughness?

P7 L249: The authors state here that the seasonal variability in surface roughness is poorly known, therefore I'm not sure they can argue that it controls the seasonal cycle in the S band (please see my previous comment).

P7 L250: I would suggest rephrasing point (iii) to make the argument the authors are trying to make clearer.

Figure 9: Should this figure have a colour scale? I would suggest a rework of this figure – it is not clear where the SP and WP zones are.

P7 L256: The authors argue here that the WP zone maximum is due to the volume

echo, but matches regions of megadunes and wind-glazed surfaces. I would appreciate if the authors addressed the following regarding this statement: (i) the Antarctic megadunes have surface features and sloped terrain on length scales similar in size to the radar footprint – how can the authors distinguish between the effects of surface and volume here? (ii) I would expect more backscatter in the summer over wind glazed regions due to the presence of large ice crystals near the surface, should that cause a backscatter peak in the summer in these regions?

P7 L264-L266: Can the authors quantify this spatial coherence, or are they implying correlation from visual inspection? Are pixels with high seasonal wind speed amplitude correlated with the winter dates of high backscatter? I'm not sure I see the relationship looking at these plots, or from the average wind speed values.

Can the authors also please expand on the seasonal amplitude of the wind speed – how is this obtained?

P8 L268: As per my previous comment, I am not sure of this correlation at Ka band either.

Figure 10: Please can the authors explicitly state the time period used in the caption. I find the elevation contours very difficult to make out, also.

P8 L282: Isn't depth-hoar predominantly formed during the late spring and summer over these wind-glazed regions, according to Scambos et al., 2012?

P8 L288: Over which time period were these grain size vertical gradients obtained? Over winter periods only or a multi-year average?

Section 5

P9 L308: "may therefore be a consequence of the presence or not of the wind-glazed areas" – I'm not sure what the authors are communicating here, I would suggest rephrasing this to make it clearer.

Technical Comments:

Please find some technical comments below, but not all I have found are listed here. In my opinion the paper is in need of a thorough proof read, with a particular focus on grammar, sentence structure and the use of more precise language to increase readability.

Title: Should read ". . .over the Antarctic Ice Sheet."

P1 L13: Please rephrase this sentence to make this clearer.

P1 L18: Should be: "At Ku band, which is intermediate. . ." and ". . .the seasonal cycle in the first zone is dominated. . ."

P1 L20: Should read ". . .should be taken into account for the more precise. . ."

P2 L60: Please rephrase this sentence for readability.

P2 L65: Please rephrase this sentence for readability.

P2 L78: "The footprint has around 5 km radius" Âň– please rephrase.

P3 L81: "To ensure post-ENVISAT mission. . ." this is incomplete, please rephrase this sentence.

P4 L148: Should this heading have a section number?

P6 L218: Please rephrase this sentence for readability.

P7 L250: ". . .interdependent and linked. . ." is a tautology, please rephrase

P8 L294: "The radar altimeter remaining on the same tracks. . ." is referring to two different satellites here, I would suggest rephrasing.

P9 L314: Should read ". . .are the key to improving. . ."

References:

[Figure]

Armitage, T.W.K. et al. (2014), Meteorological Origin of the Static Crossover Pattern Present in Low-Resolution-Mode CryoSat-2 Data Over Central Antarctica. IEEE Geoscience and Remote Sensing Letters. 11(7),pp.1295–1299.

Davis, C.H. and Ferguson, A.C. (2004), Elevation change of the Antarctic ice sheet, 1995-2000, from ERS-2 satellite radar altimetry. IEEE Transactions on Geoscience and Remote Sensing. 42(11),pp.2437–2445.

Nilsson, J., et al. (2016), Improved retrieval of land ice topography from CryoSat-2 data and its impact for volume-change estimation of the Greenland Ice Sheet, The Cryosphere, 10(6), 2953.

McMillan, M., et al. (2016), A high-resolution record of Greenland mass balance, Geophys. Res. Lett., 43, 7002–7010, doi:10.1002/2016GL069666.

---

## Referee Comment (RC2) · Anonymous Referee #2 · 12 Jan 2018

The authors have in this study analyzed seasonal variations in observed radar backscatter over the Antarctic ice sheet from two different altimetry missions spanning three different frequency bands (S, Ku, Ka). They identify two clearly marked zones over the continent exhibiting different and common frequency dependent characteristic. Exemplified, with a peak in backscattered power in the summer for the S-band, in winter for the Ka-band and for both winter and summer in the Ku-band. They attribute the difference in the observed radar backscatter to the different bands sensitivity to volume/surface scattering. To quantify the governing parameters in the snow-properties at each frequency a sensitivity study was undertaken, which took into account the snow density, grain size and snow temperature using an electromagnetic model. I find the

contribution of the paper timely and interesting, as many of these issues are not deeply looked at in altimetry. However, if find some specific sections lacking in grammar and scientific explanations.

General comments:

(1.) The font needs to be increased on all figures, currently they are to small and the text is difficult to make out. Please, also put the units of each figure inside brackets, such as "/dB" to "(dB)". Add more text to the captions that provide more explanation of what they describe, or what to look for; what should the reader look at? This helps the reader, as they do not need to go back into the manuscript looking for the associated information. I personally don't like the use of yellow in the figures, as it is hard to see sometimes, but that I will leave up to you.

(2.) The latter part of the introduction needs to be reorganized, as it jumps between altimetry missions and snowpack properties.

(3.) I would also like the boundary of the two zones to be drawn on each map to easily identify them.

(4.) Further, you say that a major limitation of the work is the lack of knowledge of the surface roughness. Have, you explored the use of ICESat for this (good overlap with Envisat)?

(5.) I think the last paragraph in the conclusion (L.310-L.316) should be re-written to more clearly state your conclusions, as I don't agree with the statement that "This as "this study mitigates". This implies that you have somehow "physically" reduce the error or corrected for it. I think its fairer to say that you have pointed to important factors that has to be considered when choosing or selecting frequency bands for new missions. Further, I would like (ii) and (iii) to be slightly more informative; how should (ii) be interpreted and why does (iii) undergo large changes etc.

Detailed comments:
L.8 "altimeter" to "altimeters"

L.9 "snowpack" to "snowpack,"

L.15 "S, Ku and Ka bands" to "different frequencies"

L.16 "Ka-band" to "Ka frequency"

L.17 "In contrast, the cycle is dominated by the surface echo at the S band" to "In contrast, at the S band, the cycle is dominated by the surface echo"

L.18 "At Ku band, which intermediate in terms of wavelength between S and Ka bands, the seasonal cycle is in the first zone dominated by the volume echo and by the surface echo in the second one" This sentence is confusing what is the first and second zone? Also, you can remove the points that Ku is between S and Ka-band as it is redundant.

L.20 You say that seasonal and spatial variations should be accounted for, but how should this be done?

L.23 Remove "within"

L.23 "of polar" to "of the polar"

L.24 "changes in volume" to "the volume change"

L.29 "distance observed" to "observed distance"

L.30 "leading" to ", leading"

L.33 "called" to "called the"

L.35 "correct" to "corrects for"

L.36 Change to: "To reduce the effect of the spatially varying radar penetration bias"...

L.37 "use" to "used"

L.37 Zwally et al (2005) used elevation residuals (crossover differences) not elevation.
L.38 As far as I know Flament et al (2012) used a linear model, solved with OLS, to estimate the sensitivity gradients. Where does the non-linear relationship come from?

L.39 "the whole" to "all"

L.42 "of" to "in the"

L.44 "of" to "of the"

L.44 "penetrating" to "interacting with"

L.45 What information on the snow pack properties does it provide?

L.50 "The ENv…" This entire section and the SARAL/Altika section should be moved down

L.60 "The radar wave…" This section should be moved to L.49

L.66 "This study is structured…" Remove this section it's redundant the reader can already understand it from the headlines.

L.80 "vertical sampling resolution" to "range gate resolution"

L.82 "25 of February" to "25th of February"

L.85 same as L.80

L.86 remove "thus"

L.87 Rewrite sentence "The frequency ..." by remove ratios

L.95 "cycles of" to "cycles of Envisats"

L.97 "cycle sigma of" to "cycle of sigma"

L.98 "fitting the time series of the observations with the following function" to "fitting the observations with the following model"

L.103 "i is the index of the along track data" Comment: This needs to be explained more

thoroughly! How large are the bins (search radius). Can you also further elaborate on how you get the number of equations in more detail.

L.104 "leading to robust inversion" Comment: How is this a robust inversion? Do you edit the data (3-sigma)? I think you mean as you only have three parameters to fit? If so just remove robust and say you solve with OLS. Further, how was the gridding performed you need to elaborate on that.

L.108 "on snow" to "of snow"

L.110 "echo and" remove echo

L.111 "been previously..." to "been previously studied by Lacroix et al (2008)"

L.112 Remove everything after Remy et al. (2015)

L.114 Rewrite first sentence to something "The snow surface can be modeled as..."

L.115 "from rough" to "from a rough"

L.116 Change to "The effective dielectric constant of the snow is"

L.117 "of snow" to "of the snow" and "and ice" to "and the ice" and "prescribed" to "modelled". Remove "statistical geometries"

L.118 "height" Comment: Use either height or elevation

L.119 put "compared to the radar wavelength" into brackets, and add "," after "coefficients" and add "a" after "from".

L.120 remove "the roughness has" and "as follows"

L.122 "at normal" to "at the normal" and add "angle" after "incident"

L.125 Remove entire sentence "When the surface snow..." it's redundant.

L.149 Remove "all"

L.150 Remove "first"

L.168 "appears in yellow" Comment: I think you should draw the boundary of the area in your figures to allow the reader to easier detect them.

L.175 When using Julian days please also provide the months inside brackets

L.195 please change "dhds" to "dh/ds" (s=sigma)

L.203 Remove "the" before "snow" and put "it is poorly known" inside brackets

L.211 "on volume" to "on the volume"

L.212 "bands" to "band levels."

L.217 "volume" to "medium"

L.217 "Along increasing" Comment: Long sentence, should be re-written.

L.218 "which increases" to "which in turn increases"

L.222 "temperature wave" maybe to "temperature gradient" and replace "to the subsurface of the" with "into". Further change "The volume echo variation" to "The variation in the volume echo"

L.224 "echo increases" Comment: Increases in what; magnitude? Make clearer!

L.239 remove "that of"

L.244 "the increase" Comment: See L.224

L.247 "which one among . . ." Sentence is worded strangely; please re-write

L.256 Remove "which" after matches, replace "greatest" with "large" and replace "of" before "radarsat" with "from".

L.269 "the distribution" to "the spatial distribution"

L.277 "in blowed" sounds strange please change sentence structure or remove.

L.283 add months after Julian days and change "By blowing" into "Persistent winds" or similar

L.288 "highest grain size vertical gradient" to "the highest vertical gradient in grain size"

L.290 "difference observed" to "observed difference"

L.294 This sentence sounds strange, maybe start something like this: "This study, using 35-day repeat radar altimetry data, allowed for..."

L.295 "used 8-year" to "used an 8-year"

L.296 "band," to "band a" and "and 3-year" to "and a 3-year"

L.297 "band all" to "band" and "covering 2002" to "covering the time period of"

L.300 remove "on the AIS", add "with a" before "maximum" and "the" before "winter"

L.302Remove "the" before "snow" and add "the seasonal changes in the" before "volume echo"

L.303 Remove "the" before "snow properties"

L.304 replace "because" with "due to" and "those properties" with "those parameters"

L.306 Remove "which is between the S and Ka bands"

L.307 "zones is" to "zones are"

L.308 "or not" to "lack of"

---

## Author Comment (AC1) · 15 Feb 2018

In this study, Adodo et al. observe seasonal variations in radar backscatter over the Antarctic Ice Sheet using three different radar frequencies. The authors define regions over Antarctica where backscattered power is found to peak in the summer for the S band, winter for the Ka band, and in both summer and winter for the intermediate Ku band. The authors perform a sensitivity study to help understand the effects of surface snow density, snow temperature, and snow grain size on backscattered power from

each radar band. This study, in particular the delineation of these summer and winter 'peak zones', as referred to by the authors, represents a worthwhile addition to the literature. However, in my opinion some of the reasoning the authors provide in the discussion section to relate these seasonal variations to physical processes is lacking in places, and I would appreciate if they could respond to the following comments.

• We appreciate that the reviewer considers our study as "a worthwhile addition to the literature", and we would like to thank her/him for her/his careful and thorough reading of this manuscript, and comments (shown in italics). Our responses are given below.

P1 L8: I would suggest rephrasing to "radar wave interaction with the snow. . ." instead of "radar wave penetration. . .".

• The sentence has been modified as suggested. The sentence now reads (Line 8) :

" The radar wave interaction with the snow provides information both on the surface and the subsurface of the snowpack, due to its dependence on the snow properties."

P1 L27: I feel the phrase "More or less corrected" is quite vague here: corrections for atmosphere/ionosphere and slope errors are well established in the literature and minimise these errors with good accuracy – radar wave penetration is the main outstanding problem listed.

• The text has been modified according to the reviewer's suggestion. Indeed, in the text we have separated the first errors and the last the one that is more critical. The text has been rephrased as follows (Line 26 to 31) :

"However, altimetric observations are affected by several errors : errors due to atmospheric/ionospheric propagations, slope error and error due to the radar wave penetration into the cold and dry snow (Ridley and Partington, 1988). The first two errors are usually corrected with good accuracy (Remy et al., 2012; Nilsson et al., 2016), while

the last one is the most critical and the most challenging problem to tackle (Remy et al., 2012) as it results in an overestimation of the observed distance between the satellite and the target, leading to a negative bias in the surface elevation estimation."

P1 L30: I have some concerns with the 2012 Greenland melt event being used as an example here. This is a positive elevation bias (not negative as the authors discuss in the preceding sentence) caused by a resetting of the radar scattering horizon due to an anomalous surface melt event. This is a process without equivalent in Antarctica, and one that has also been corrected for in the literature when measuring surface elevation change with radar altimetry (Nilsson et al., 2016, McMillan et al., 2016). In my opinion the authors should clarify this here, or include more examples on the effects of radar penetration on time series of elevation in Antarctica from radar altimetry in order to better establish the problem they are addressing.

⇢ The Greenland example has been removed and the text has been edited. We have added two references. The sentence now reads (Line 31 to 35) :

"The magnitude of the penetration error on the estimated surface elevation is between a few tens of centimeters and few meters (Remy and Parouty, 2009). For instance, Michel et al. (2014) have found a surface elevation difference of -0.5 ăm between ENVISat and ICESat crossover points over Antarctica. Authors relate this negative bias to the difference in the penetration depth between the radar altimeter wave that penetrates within the snowpack and the laser altimeter beam that not penetrates within the snowpack."

"Michel, A., Flament, T. and Remy, F.: Study of the Penetration Bias of ENVISAT Altimeter Observations over Antarctica in Comparison to ICESat Observations, Remote Sens., 6(10), 9412–9434, doi:10.3390/rs6109412, 2014."

P1 L41: The authors fail to mention the work of Davis and Ferguson here (Davis and Ferguson, 2004), however I feel this is a significant contribution to the literature which the authors should include.

• Our impression is that the improved method of Davis and Ferguson (2004) is original but does not fit with the scope of this paper. Our purpose is to mention the two most popular approaches that process the satellite data using the crossover or the along-track analysis, and use the backscattering coefficient to adjust and reduce the effect of the spatially varying radar penetration error. Note that, we are not proposing a new method but addressing the contrasted behavior of the seasonal variations in the observed radar backscatter between frequencies and regions.

P2 L46: The authors should cite Ridley and Partington 1988 here.

• Thank you for spotting this lack. The reference has been cited.

P2 L50: I think it would be helpful for the reader to know the wavelength of each radar band in addition to the frequency, either here on in Section 2.1.

• The wavelength of each radar band has been added.

" S (3.2 GHz $\sim$ 9.4 cm), Ku (13.6 GHz $\sim$ 2.3 cm) and Ka (37 GHz $\sim$ 0.8 cm) bands."

P1 L55: The manuscript states later on that the orbit of AltiKa has recently been shifted so I would suggest amending this sentence to reflect this.

• The sentence is reworded to account for reviewer's suggestion. The sentence now reads (Line 69):

" The launch in March 2013 of the radar altimeter SARAL/Altika that operates at the Ka band (37 GHz $\sim$ 0.8 cm) and had the same 35-day phased orbit as ENVISAT until March 2016, allowed comparisons with much higher frequencies for the first time."

P2 L79: The authors describe the radar waveform, but the concept has already been introduced in the previous section. I would suggest formally defining the waveform where it is first mentioned as opposed to here, as it is a key concept needed for the paper.

[Figure]

• We have moved the waveform definition to the appropriate paragraph in the Introduction (Line 38).

P3 L87: The authors should rephrase this sentence to make it more clear that the ICE-2 retracker is used to obtain backscattering coefficients for the Ku and S bands.

• The text has been revised, as suggested. The sentence now reads (Line 87):

"The ICE-2 retracking process was applied to the Ka, Ku and S band waveforms allowing estimation of the range, the backscattering coefficient ($\sigma$0), the leading edge width and the trailing edge slope."

Do the authors consider ascending and descending tracks separately? A previous study has shown radar backscatter has an anisotropic dependence resulting from the interaction between the radar polarization direction and wind induced features of the firn (Armitage et al., 2013).

• This point has not been addressed in the manuscript because we found no influence on time varying component. In fact, we explored the seasonal amplitude and phase of the backscattering coefficient using crossover and along tracks analysis. We considered the ascending and descending passes separately at the satellite cross tracks and at the along-track. No significant difference or geographical pattern (similar to that observed by Remy et al. (2012) or Armitage et al. (2014)) have been found. We found that the azimuthal anisotropic effect is quite stationary from one cycle to another, therefore does not affect the seasonal characteristics. Consequently our analysis of the seasonal cycle of the backscattering coefficient using both ascending and descending passes at along tracks analysis is free of anisotropic effects. In order to keep a higher density of available data points and cover most Antarctic Ice sheet, we have prioritized the use of both the ascending and descending passes instead of one of them.

• A sentence has been added to specify this point in the section 2.2. (Line 108) :

"We have found that along track analysis of the seasonal parameters of $\sigma 0$ showed no dependence to anisotropic effects. In the following, both ascending and descending measurements are mixed to keep a high density of observations and cover most AIS ($\sim$ 1.9 million data points)."

Figure 1: Can the authors please include a map to indicate where this location is in relation to the continent, and along the orbit tracks of the 3 bands.

• The illustrated location has been indicated in the maps with a cross mark.

P3 L105: Do the authors place any controls on poorly constrained fits due to e.g. poor match between observed and modelled seasonal peaks?

• We placed a criterion on the length of the time series. The sentence has been edited to reflect this missing information. The sentence now reads (Line 105) :

"The fit was done with the Ordinary Least Squares (OLS) method and all data points with time-series length less than 11 cycles (about a year) were discarded."

• The OLS method fits data with the minimized root mean square error (rmse). The minimum rmse of the fit using equation 1 are under 1ÂădB at the three frequencies and the regions with highest rmse are near the coasts, on ice-shelfs and on a part of the Western Antarctic. This may be explained by the large variation in the signal in these regions linked to ocean influence. In East Antarctica, where the delineation is the most remarkable, the rmse is less than 0.5ÂădB in the inland of the continent. We have confidence in the OLS method because we also checked manually the fit at many data points.

In addition, can the authors please provide more information on how the amplitude and phase are gridded. Do they use the mean? If so, are there grid cells which have a high variance? How many coefficients, on average, are binned into a 5 km grid cell for each radar band? What data coverage does this provide in more challenging regions such as the margins and the Peninsula?

• We gridded data only for visualization needs as explained in the text. As suggested by the reviewer, we have added a text to explain thoroughly how the data are gridded over the AIS (Line 110) :

"For visualization needs, seasonal parameters are interpolated on a map of grid by averaging with Gaussian weights. We considered all data points within a 25 km radius and weighted with a decorrelation radius of 10 km. "

• A text has been added to detail the dataset used in the section 2.1 (Line 91) :

"The ENVISAT and AltiKa datasets used in this study were averaged at a 1-km scale on the ENVISAT nominal orbit."

P4 L151: Can the authors please comment on the validity of applying this firn density profile obtained at one location to the rest of the Antarctic ice sheet – how sensitive are the results to this assumption?

• The assumption of the firn density profile used has a negligible effect on the results of the simulations. The simulations show the same evolution in the magnitude when applying a constant vertical density profile. The snow density profile reliability will be questionable if the absolute value of the backscattering coefficient had been simulated. In fact, snow density profile ideally increases with depth, due to snow compaction over time. One can idealize the firn density profile over the AIS with a given density profile for a sensitivity tests, as snow density variation range is known.

A text has been added to specify this :

"The choice of the vertical density profile has a negligible effect on the results of the sensitivity test."

P5 L168: Which day/month of the year do these peaks correspond to? More information can be provided to the reader here.

• Details on day/month of the year have been added in brackets, as also suggested

by the reviewer 2. The summer and winter seasons were accurately defined further in the section (Line 179).

Figure 2: It would be useful to plot elevation contours (or an inset elevation map) if elevation is used to delineate backscatter patterns in the text. It would also be helpful for the authors to indicate the locations of regions they refer to in the text (e.g. Wilkes Land, Dronning Maud Land).

• Elevation was not used at all in the manuscript. Instead the delineation of the radar backscatter patterns is the 'date at which the backscatter reaches a maximum' derived from the seasonal phase of the backscattering coefficient at the Ku frequency.

Would it also be possible for the authors to mark out the summer and winter peak zones they define in the text? Finally, I would also suggest using a different colour scale, the differences between pale yellow-green-blue are quite hard to make out.

• As suggested by both reviewers, the boundaries of the SP zone have been drawn on each map. These boundaries show regions where the backscattering coefficient at the Ku band peaks before April. We have also changed the color scale and have plotted a cross mark on the map to indicate the location of the time series shown in figure 1.

P5 L177: Do these percentages refer to the observed area, or the entire Antarctic ice sheet? This applies to any percentage stated in this way.

• The correction has been made. These percentages refer to the observed area. The sentence now reads (Line 181) :

"With these definitions, the WP and SP zones represent 42% and 45% of the observed area, respectively."

P5 L178: Do the percentages in brackets also refer to the area of these summer and winter zones for each band? As written it is not clear to me, I would suggest rephrasing this.

• As suggested, we have rephrased the sentence as follow (Line 182) :

"The histogram of the date of maximum $\sigma 0$ at the S and Ka bands are unimodal with a peak in summer for a lower frequency (WP : 11%, SP : 66%, using the summer and winter periods previously defined) and a peak in winter for a higher frequency (WP : 50%, SP : 14%, using the summer and winter periods previously defined)."

Figure 4: I would suggest using a different colour scale which is preferably divergent to make the figure clearer.

• The suggestion has been taken into account.

P6 L187: How are the uncertainties in backscatter coefficient derived here? They appear to be quite large to me.

• If we understand correctly this comment, we have not derived the uncertainties of the backscatter coefficient but the average and the standard deviation of the seasonal amplitude in the WP zone, using all the data points where the backscattering coefficient peaks between the Julian days 175 and 275 (June to September) , i.e. around 42% of the observations (see Figure 6).

P6 L195: Can the authors please expand on how they are deriving surface elevation, is it also from the ICE-2 retracker, and binned at the same 5 km grid used for the backscatter coefficients? Have these elevations been corrected for atmosphere/slope? In addition, how are the values of dhd$\sigma$ derived?

• The surface elevation were indirectly estimated from the retracked range (computed with the ICE-2 retracker) at each data point. The surface elevation has been corrected for atmospheric errors. However, there is no need to correct for slope error because the computation is done at each along track data point. Seasonal parameters have been gridded for visualization needs.

• dh and d$\sigma 0$ are the elevation residuals and the backscattering coefficient residuals, respectively, obtained by subtracting the mean signal from the times series.

 c dh/d$\sigma$0 is the correlation gradient or the slope of the linear relationship between residuals of the elevation and backscattering coefficient at each available data points over the AIS.

The text has been expanded as follow (Line 201 to 212).

"Figure 7 shows the spatial distribution of temporal variations of the estimated surface elevation residuals with respect to $\sigma$0 residuals at the Ku band, hereafter denoted dh/d$\sigma$0. The surface elevation were indirectly estimated from the retracked range (computed with the ICE-2 retracker) at each data point and were corrected for atmospheric errors. dh and d$\sigma$0 were derived by subtracting the mean value from the time series of the elevation and backscatter, respectively. dh/d$\sigma$0 represents the correlation gradient or the slope at each data points over the AIS. Negative values of dh/d$\sigma$0 indicate that surface elevation decreases when $\sigma$0 increases, implying that temporal variations in $\sigma$0 are due to changes in the deep snowpack properties, i.e. in the volume echo. In fact, the inverse relationship between surface elevation and $\sigma$0 is related to a greater backscatter from depth that shifts more power to greater delay times in the received waveform, thus increasing the retracked range and decreasing the estimated elevation (Armitage et al., 2014). On the contrary, positive values of dh/d$\sigma$0 indicate that the surface elevation increases with $\sigma$0. In this case, the temporal variations of $\sigma$0 are related to changes in the surface echo. The map in Fig. 7 shows that near-zeros and negative values of dh/d$\sigma$0 (in blue) are found in the WP zone. This means that the WP zone undergoes large variations of volume echo."

Figure 7: The units in the caption state dB not m/dB.

 c The correction has been done.

P6 L217: "...resulting in a decreases of the radar wave in the volume. . .". Are the authors referring to a decrease of backscattered power? I suggest the use of more precise language in instances like this. Also decreases should be decrease.

• The correction has been done. We are referring to the backscattered power. The sentence now reads (Line 228): "This sensitivity is explained by the fact that increasing snow temperature increases absorption resulting in a decrease of the radar wave penetration in the medium, thus limiting the volume echo."

P6 L220: It would be helpful to show this WP zone on Figure 4 to make clear to the reader.

• The suggestion has been taken into account.

P7 L222: Do the authors have any evidence to back up this assertion of volume echo variations being driven by temperature? I agree this is a reasonable conclusion to propose, however the authors do not offer enough evidence to convince that this is indeed the case. I would suggest that the sentence is reworded to make the authors argument clearer.

• In Figure 7, we showed that a greater backscatter comes from depth in the WP zones where the signal peaks in the winter at the Ku and Ka band. In addition, in Figure 8 we demonstrated that the variations in the volume scattering with respect to the snow temperature is 2 times greater than that of the density and grain size. In Figure 3, there is a lag of about 40 days between the peaks of the Ka and Ku bands in the WP zone (see, Figure 4). We have simulated the seasonal phase of the volume echo (Figure not shown) and found that only the temperature gradient can cause a lag between the Ku and Ka bands.

The sentence has been reformulated as follow (Line 234):

"As the temperature controls the snow grain metamorphism and the radar wave penetration depth, the variation in the volume echo would be predominantly driven by the seasonal variations of snow temperature in the WP zone."

P7 L230-240: I am not sure I agree with the soil analogy – in my opinion it doesn't offer any clarity to the reader and isn't needed. Can the authors please expand on what they

mean when they state the snow surface is sensed "as a volume scattering medium at the Ka band" – in reality there will always be a surface component of the radar echo controlled by incidence angle and topography on the footprint scale.

• We can understand the concern of the reviewer about the soil analogy. Here, we are not comparing the soil and snow media but their common surface scattering behaviors in radar altimeters and in surface scattering models. It is important to keep in mind that we are addressing, in this study, the seasonal characteristics of the observed backscattered power. It is obvious that a surface component is always present in the signal, but if it does not vary over time, it can not explain the seasonal cycle. For instance, the spatial distribution of the seasonal amplitude of the Ka band is an evidence that the surface component is present and would be much greater in the wind-glazed surfaces region (smooth and polish surface).

This paragraph has been edited to explain thoroughly our argumentation and the soil analogy has been removed. Now reads :

"From the S to Ka band, the radar wavelength decreases by a factor 12 from 9.4 cm to 0.8 cm corresponding to a scale change from centimeter to millimeter. The scale at which the surface roughness plays a role in radar backscattering coefficient depend on the radar wavelength (Ulaby et al., 1982). On a rough surface, the surface scattering consists of two components: the coherent and incoherent scattering (Ulaby et al., 1982). The former is the scattered component in the specular direction while the latter is the scattered component in all directions. As the radar wavelength is shortened to less than a centimeter, the surface appears rougher and the surface coherent component vanishes (Ulaby et al., 1982). The surface incoherent component magnitude is small, and thus is concealed by the volume scattering which consists of only incoherent scattering. The backscattering coefficient at a smaller wavelength or on a rougher surface would be consisted of only incoherent components therefore appears as a volume-scattering medium. Simulations in Fig. 7 emphasize this contention showing a greater amplitude of the volume echo at a higher frequencies. We can therefore

argue that the seasonal cycle of the observed $\sigma 0$ at the Ka band is governed by the volume echo. This explains the peak of the observed $\sigma 0$ in the winter at the Ka band over the AIS."

P7 L244: Do the authors mean to reference Fig. 8 here and not Fig. 3?

âǍć We are referring to the histograms of the date of maximum $\sigma 0$ in Fig. 3.

P7 L246: Do the authors have any evidence for a seasonal cycle of snow surface roughness?

âǍć We have demonstrated that the volume echo is nearly constant at S band therefore can not explain the observed seasonal cycle. In the other hand, we have demonstrated that the seasonal cycle observed at the three frequencies can not be explained solely by the snow density changes. So, the only remaining reasonable conclusion is that the S band seasonal cycle stems from the surfaces roughness seasonal variations. Nevertheless, we do not have clear and consistent evidence of widespread seasonality in surface roughness.

A text has been added to clarify our argumentation (Line 260) :

"Therefore, it is likely that the seasonal cycle of the observed $\sigma 0$ at the S band, predominantly driven by the surface echo, stems from the seasonal cycle of the snow surface roughness. There is no field observation that confirms this fact, but our findings suggest that such information would help to understand the altimetric signal in the future."

P7 L249: The authors state here that the seasonal variability in surface roughness is poorly known, therefore I'm not sure they can argue that it controls the seasonal cycle in the S band (please see my previous comment).

âǍć We have demonstrated that neither the volume echo nor surface snow density can explain the seasonal signal observed at S band. Since, the seasonal cycle observed at S band can only be explained by the surface echo which is related to the surface

roughness and density, one can inferred the seasonal cycle at S band to be driven by the surface roughness (please see our previous answer to this question).

P7 L250: I would suggest rephrasing point (iii) to make the argument the authors are trying to make clearer.

• We have changed the text to (Line 264) :

"(iii) the relation between the surface snow roughness and density is complex because both variables are interdependent. The denser the snow surface, the larger the effect of surface roughness is. This amplification is due to the increase of the effective dielectric discontinuity with density (Fung, 1994)."

Figure 9: Should this figure have a colour scale? I would suggest a rework of this figure – it is not clear where the SP and WP zones are.

• This map is a RADARSAT mosaic which is most objectively shown with gray scale. The rework of the previous figures should now make this one clearer.

P7 L256: The authors argue here that the WP zone maximum is due to the volume echo, but matches regions of megadunes and wind-glazed surfaces. I would appreciate if the authors addressed the following regarding this statement: (i) the Antarctic megadunes have surface features and sloped terrain on length scales similar in size to the radar footprint – how can the authors distinguish between the effects of surface and volume here? (ii) I would expect more backscatter in the summer over wind glazed regions due to the presence of large ice crystals near the surface, should that cause a backscatter peak in the summer in these regions?

• Figure 4 shows that a greater backscatter comes from at depth in the WP zone. This means that the volume contribution is more important in the WP zones (see Armitage et al., 2014). Also, the presence of wind-glazed surfaces in these regions indicates that the surface varies very little over time due to the lack of snow accumulation related to the strong and persistent winds in the regions. Therefore, even if the surface

component may be higher, the seasonal variations can only be ascribed to the volume component.

• There will be more backscatter in the summer in these regions if only the volume echo was predominantly driven by the snow grain seasonal cycle. This is not the case. As we asserted on line 235 : The volume echo would be mainly driven by the snow temperature.

P7 L264-L266: Can the authors quantify this spatial coherence, or are they implying correlation from visual inspection? Are pixels with high seasonal wind speed amplitude correlated with the winter dates of high backscatter? I'm not sure I see the relationship looking at these plots, or from the average wind speed values.

• Yes, we are implying correlation from visual inspection. With the rework of the figures, this visual coherence is hopefully more obvious and clear. Moreover, the computed correlation coefficient between the date of maximum of backscatter and the seasonal wind speed amplitude, after the interpolation of the date of maximum backscatter at Ku band on a 25 km grid cell (same to that of the wind speed dataset) by averaging with Gaussian weights considering all data points within a 25 km radius and weighting with a decorrelation radius of 10 km, is r = 0.4 (p<0.01).

• The figure 1 shows the mean (blue) and median (red) seasonal wind speed amplitude with respect to the date of maximum backscatter at the Ku band. One can observe that high seasonal wind speed amplitude is correlated with the winter dates of high backscatter at the Ku band (between Julian days 175 and 275 (June to September)).

Can the authors also please expand on the seasonal amplitude of the wind speed – how is this obtained?

• The correction has been done. We have added this details (Line 278) :

"To further investigate this point, we used ERA-Interim reanalysis wind speed data

supplied by ECMWF (European Centre For Medium-Range Weather Forecasts) on the period 2002 to 2010, corresponding to that of the Ku band. Equation 1 is used to compute the seasonal characteristics of the wind speed by replacing $\sigma0$ with the wind speed. A visual inspection shows a high spatial coherence of the seasonal amplitude of the wind speed (Fig.Âă10a) patterns with the date of maximum $\sigma0$ over the seasonal cycle at the Ku band (Fig.Âă2b)."

P8 L268: As per my previous comment, I am not sure of this correlation at Ka band either.

  The rework of the figures should make this point clearer.

Figure 10: Please can the authors explicitly state the time period used in the caption. I find the elevation contours very difficult to make out, also.

  We have used the time period of ENVISAT for the wind speed data (2002-2010). The caption has been modified to take into account the reviewer suggestion. We have deleted the elevations contours because, they are not necessary.

The caption now reads :

"Figure 10 : Seasonal wind speed amplitude (left) and average (right). Data are extracted from ERA-Interim reanalysis provided by ECMWF on a km2 grid cells, on the periods 2002 to 2010 corresponding to that of ENVISAT lifetime. Black contour lines delineate regions where the backscattering coefficient at the Ku band peaks before April. The star mark shows the location of the time series plotted in Figure 1. No observations are available beyond 81.5°ÂăS (black dotted circle)."

P8 L282: Isn't depth-hoar predominantly formed during the late spring and summer over these wind-glazed regions, according to Scambos et al., 2012?

  Scambos et al. (2012) have conducted the in situ measurements the late spring and summer and observed a depth hoar formation caused by the sun light penetration in the smooth and polished surfaces of the snowpack. Champollion et al. (2013) have

suggested a depth hoar formation at Dome C caused by a strong temperature gradient (positive) in the snowpack during the winter. Since there is a strong temperature gradient in the WP regions in the winter, depth hoar will develop.

"Champollion, N., Picard, G., Arnaud, L., Lefebvre, E. and Fily, M.: Hoar crystal development and disappearance at Dome C, Antarctica: observation by near-infrared photography and passive microwave satellite, The Cryosphere, 7(4), 1247, 2013."

P8 L288: Over which time period were these grain size vertical gradients obtained? Over winter periods only or a multi-year average?

• These grain size vertical gradients were obtained over a multiyear average from 1987 to 2002. The sentence now reads (Line 305) :

"For instance, Brucker et al. (2010) have found the highest vertical gradient in grain size, obtained over a multiyear average from 1987 to 2002, in the regions of the WP zone."

P9 L308: "may therefore be a consequence of the presence or not of the wind-glazed areas" – I'm not sure what the authors are communicating here, I would suggest rephrasing this to make it clearer.

• The correction has been done. The sentence now reads (Line 324 ) :

"The geographical patterns of the WP and SP zones are related to the seasonal amplitude of the wind speed. This is a result of the presence or lack of wind-glazed surfaces, induced by strong and persistent winds in the megadune areas. "

Technical Comments:

Please find some technical comments below, but not all I have found are listed here. In my opinion the paper is in need of a thorough proof read, with a particular focus on grammar, sentence structure and the use of more precise language to increase readability.

[Figure]

• Following the both reviewers technical comments, we have added and rephrased numerous sentences in the revised manuscript for readability. A special attention has been given to double-checking the english grammar, proofreading and sentence structure.

Title: Should read ". . .over the Antarctic Ice Sheet."

• As suggested by the reviewer, we have corrected the title.

P1 L13: Please rephrase this sentence to make this clearer.

• We have rephrased the sentence as follow in the abstract (Line 12) :

"We identified that the backscattering coefficient at Ku band reaches a maximum in winter in part of the continent (Region 1) and in the summer in the remaining (Region 2) while the evolution at other frequencies is uniform."

P1 L18: Should be: "At Ku band, which is intermediate. . ." and ". . .the seasonal cycle in the first zone is dominated. . ."

• The correction has been made.

P1 L20: Should read ". . .should be taken into account for the more precise. . ."

• The correction has been made.

P2 L60: Please rephrase this sentence for readability.

• We have edited the sentence. The sentence now reads (Line 58) :

"The radar wave interaction with snow provides information on the snowpack surface and sub-surface properties, but it complicates the altimetric signal interpretation because the latter would be sensitive to many more snow parameters than if the signal only comes from the surface."

P2 L65: Please rephrase this sentence for readability.

• We have rephrased the sentence. The sentence now reads (Line 63) :

"The aim of this paper is to determine the prevailing snow parameters that drive the seasonal cycle of the observed backscattering coefficient at different radar frequencies and locations over the AIS."

P2 L78: "The footprint has around 5 km radius" Ân– please rephrase.

• The sentence has been rephrased. The sentence now reads (Line 80):

"the satellite footprint has typically a 5 ăkm radius and no data were acquired above 81.5°ĂăS due to its orbit maximum inclination."

P3 L81: "To ensure post-ENVISAT mission. . ." this is incomplete, please rephrase this sentence.

• The sentence has been completed as follow (Line 82) :

"To ensure a long and homogeneous time series with post-ENVISAT missions and to complement the Ocean Surface Topography Mission (OSTM)/Jason (Steunou et al., 2015), the Satellite for ARgos and ALtiKa (SARAL)/AltiKa was launched on 25th February, 2013, by a joint CNES-ISRO (Centre National d'Etudes Spatiales - Indian Space Research Organisation) mission, on the same 35-day repeat cycle orbit as ENVISAT."

P4 L148: Should this heading have a section number?

• No, this heading has not a section number.

P6 L218: Please rephrase this sentence for readability.

• The correction has been done. The sentence now reads (Line 227) :

"This sensitivity is explained by the fact that increasing snow temperature increases absorption resulting in a decrease of the radar wave penetration in the medium, thus limiting the volume echo."

P7 L250: ". . .interdependent and linked. . ." is a tautology, please rephrase

• The correction has been made, and the redundant word has been removed. The sentence now reads (Line 264) :

"(iii) the relation between the surface snow roughness and density is complex because they are interdependent. The denser the snow surface, the larger the effect of surface roughness is. This amplification is due to the increase of the effective dielectric discontinuity with density (Fung, 1994)."

P8 L294: "The radar altimeter remaining on the same tracks. . ." is referring to two different satellites here, I would suggest rephrasing.

• The sentence has been rephrased as follow (Line 311) :

" This study, using 35-day repeat radar altimetry data, allowed to carry out this spatial and temporal comparatives analysis of the seasonal amplitude and date of maximum $\sigma 0$ at the S, Ku and Ka bands."

P9 L314: Should read ". . .are the key to improving. . ."

• The correction has been done.

References:

Armitage, T.W.K. et al. (2014), Meteorological Origin of the Static Crossover Pattern Present in Low-Resolution-Mode CryoSat-2 Data Over Central Antarctica. IEEE Geoscience and Remote Sensing Letters. 11(7),pp.1295–1299.

• This reference has been cited.

Davis, C.H. and Ferguson, A.C. (2004), Elevation change of the Antarctic ice sheet, 1995-2000, from ERS-2 satellite radar altimetry. IEEE Transactions on Geoscience and Remote Sensing. 42(11),pp.2437–2445.

• This reference does not fit with the scope of this paper.

Nilsson, J., et al. (2016), Improved retrieval of land ice topography from CryoSat-2

data and its impact for volume-change estimation of the Greenland Ice Sheet, The Cryosphere, 10(6), 2953.

• This reference is already cited in the original text.

McMillan, M., et al. (2016), A high-resolution record of Greenland mass balance, Geophys. Res. Lett., 43, 7002–7010, doi:10.1002/2016GL069666.

• This reference has not been cited.

[Figure]

[Figure]

**Fig. 1.** Histogram of wind speed seasonal amplitude with respect to the Ku band backscattered coefficient date of maximum

[Figure]

**Fig. 2.** examples of the new color scale, and boundaries of WP and SP

---

## Author Comment (AC2) · 15 Feb 2018

The authors have in this study analyzed seasonal variations in observed radar backscatter over the Antarctic ice sheet from two different altimetry missions spanning three different frequency bands (S, Ku, Ka). They identify two clearly marked zones over the continent exhibiting different and common frequency dependent characteristic. Exemplified, with a peak in backscattered power in the summer for the S-band, in winter for the Ka-band and for both winter and summer in the Ku-band. They attribute

the difference in the observed radar backscatter to the different bands sensitivity to volume/surface scattering. To quantify the governing parameters in the snow-properties at each frequency a sensitivity study was undertaken, which took into account the snow density, grain size and snow temperature using an electromagnetic model. I find the contribution of the paper timely and interesting, as many of these issues are not deeply looked at in altimetry. However, if find some specific sections lacking in grammar and scientific explanations.

• We thank the reviewer for constructive and suggestions (shown in italics). Our responses are given below. As suggested, we have reviewed carefully the entire manuscript and removed redundancies, we have corrected grammatical issues and have improved the clarity of the figures as shown in the revised manuscript. General comments:

(1.) The font needs to be increased on all figures, currently they are to small and the text is difficult to make out. Please, also put the units of each figure inside brackets, such as "/dB" to "(dB)". Add more text to the captions that provide more explanation of what they describe, or what to look for; what should the reader look at? This helps the reader, as they do not need to go back into the manuscript looking for the associated information. I personally don't like the use of yellow in the figures, as it is hard to see sometimes, but that I will leave up to you.

• We have changed the color scale in each figure and provide more information in the captions.

(2.) The latter part of the introduction needs to be reorganized, as it jumps between altimetry missions and snowpack properties.

• As suggested by the reviewer in the technical comments, we have reorganized the introduction to correct for this problem.

(3.) I would also like the boundary of the two zones to be drawn on each map to easily

identify them.

• As suggested by both reviewers, the boundaries of the SP zone have been drawn on each map. These boundaries show regions where the backscattering coefficient at the Ku band peaks before April. We have also changed the color scale and have plotted a cross mark on the map to indicate the location of data point shown in figure 1.

(4.) Further, you say that a major limitation of the work is the lack of knowledge of the surface roughness. Have, you explored the use of ICESat for this (good overlap with Envisat)?

• We did not closely look at the ICESat data. The radar backscatter is related to the surface roughness at the scale of the radar wavelength (Ulaby et al., 1982) and larger. Here, radar wavelength are less than tens of centimeters (0.8 cm, 2.3 cm and 9.4 cm for Ka, Ku and S bands, respectively) while ICEsat provides only surface elevation profiles at the metric-to-kilometric scales. This prevents us to use such surface roughness.

(5.) I think the last paragraph in the conclusion (L.310-L.316) should be re-written to more clearly state your conclusions, as I don't agree with the statement that "This as "this study mitigates". This implies that you have somehow "physically" reduce the error or corrected for it. I think its fairer to say that you have pointed to important factors that has to be considered when choosing or selecting frequency bands for new missions. Further, I would like (ii) and (iii) to be slightly more informative; how should (ii) be interpreted and why does (iii) undergo large changes etc.

• As suggested by the reviewer, we have rewritten the last paragraph of the conclusion. The (ii) and (iii) have been removed and we have specified what have been done. Now reads (Line 327) :

"This investigation provides new information on the Antarctic Ice Sheet surface seasonal dynamics and provides new clues to build robust correction of the altimetric

surface elevation signal. Multi-frequency sensors are the key to improving the understanding of the physics of radar altimeter measurements over the AIS. An important limitation of this study is the lack of information on the seasonal variability of the snow surface roughness in Antarctica, which will be the topic of future work."

Detailed comments:

L.8 "altimeter" to "altimeters"

• The correction has been made.

L.9 "snowpack" to "snowpack,"

• The correction has been made.

L.15 "S, Ku and Ka bands" to "different frequencies"

• The correction has been made

L.16 "Ka-band" to "Ka frequency"

• The correction has been made.

L.17 "In contrast, the cycle is dominated by the surface echo at the S band" to "In contrast, at the S band, the cycle is dominated by the surface echo"

• The correction has been made. The sentence now reads (Line 19):

"In contrast, at S band, the cycle is dominated by the surface echo."

L.18 "At Ku band, which intermediate in terms of wavelength between S and Ka bands, the seasonal cycle is in the first zone dominated by the volume echo and by the surface echo in the second one" This sentence is confusing what is the first and second zone? Also, you can remove the points that Ku is between S and Ka-band as it is redundant.

• We have corrected for this confusion and added more information on the zones. The sentence now reads (Line 13):

"We identified that the backscattering coefficient at Ku band reaches a maximum in winter in part of the continent (Region 1) and in the summer in the remaining (Region 2) while the evolution at other frequencies is relatively uniform over the whole continent."

"At Ku band, the seasonal cycle is in the Region 1 dominated by the volume echo and by the surface echo in the other one."

L.20 You say that seasonal and spatial variations should be accounted for, but how should this be done?

• The corrections suggested by the reviewer on L15 to L20 have been made and the text reworded as follow:

"We identified that the backscattering coefficient at Ku band reaches a maximum in winter in part of the continent (Region 1) and in the summer in the remaining (Region 2) while the evolution at other frequencies is relatively uniform over the whole continent. To explain this contrasted behavior between frequencies and between regions, we studied the sensitivity of the backscattering coefficient at three frequencies to several parameters (surface snow density, snow temperature and snow grain size) using an electromagnetic model. The results show that the seasonal cycle of the backscattering coefficient at Ka frequency, is dominated by the volume echo and is mainly driven by snow temperature evolution everywhere. In contrast, at S band, the cycle is dominated by the surface echo. At Ku band, the seasonal cycle is in the Region 1 dominated by the volume echo and by the surface echo in the other one. This investigation provides new information on the seasonal dynamics of the Antarctic Ice Sheet surface and provides new clues to build more accurate correction of the radar altimetric surface elevation signal in the future."

L.23 Remove "within"

• The correction has been done.

L.23 "of polar" to "of the polar"

  The correction has been done.

L.24 "changes in volume" to "the volume change"

  The correction has been done.

L.29 "distance observed" to "observed distance"

  The correction has been done

L.30 "leading" to ", leading"

  The correction has been done.

L.33 "called" to "called the"

  The correction has been done.

L.35 "correct" to "corrects for"

  The correction has been done.

L.36 Change to: "To reduce the effect of the spatially varying radar penetration bias". . .

  The correction has been done.

L.37 "use" to "used"

  The correction has been done.

L.37 Zwally et al (2005) used elevation residuals (crossover differences) not elevation.

  The correction has been done.

L.38 As far as I know Flament et al (2012) used a linear model, solved with OLS, to estimate the sensitivity gradients. Where does the non-linear relationship come from?

  Yes, we agree Flament et al. (2012) have really used a linear model not non-linear

as we have written. We have corrected this.

L.39 "the whole" to "all"

• The correction has been done.

L.42 "of" to "in the"

• The correction has been done.

L.44 "of" to "of the"

• The correction has been done.

L.44 "penetrating" to "interacting with"

• The correction has been done.

L.45 What information on the snow pack properties does it provide?

• The following snowpack properties have been cited in bracket on Line 52 :

"(surface roughness and density, temperature, grain size, and stratification)".

L.50 "The ENv. . ." This entire section and the SARAL/Altika section should be moved down

• We have moved the paragraph to a more appropriate place in the introduction on Line 65.

L.60 "The radar wave. . ." This section should be moved to L.49

• We have moved the paragraph to a more appropriate place in the Introduction on Line 58.

L.66 "This study is structured. . ." Remove this section it's redundant the reader can already understand it from the headlines.

• As suggested by the reviewer we have removed the corresponding text.

L.80 "vertical sampling resolution" to "range gate resolution"

• The correction has been done .

L.82 "25 of February" to "25th of February"

• The correction has been done.

L.85 same as L.80

• The correction has been done.

L.86 remove "thus"

• The correction has been done.

L.87 Rewrite sentence "The frequency ..." by remove ratios

• The sentence has been reworded as follow on line 89 :

" The difference between the Ka and Ku bands, and the Ka and S band are up to a factor of 2.7 and 11.6, respectively, which results in different sensitivity to the surface and the subsurface characteristics."

L.95 "cycles of" to "cycles of Envisats"

• The correction has been done.

L.97 "cycle sigma of" to "cycle of sigma"

• The correction has been done.

L.98 "fitting the time series of the observations with the following function" to "fitting the observations with the following model"

• The correction has been done.

L.103 "i is the index of the along track data" Comment: This needs to be explained more thoroughly! How large are the bins (search radius). Can you also further elaborate on

how you get the number of equations in more detail.

• The text was incomplete. The previous corrections and suggestions from the first reviewer will make this sentence more clearer and informative. The sentence now reads on line 105:

"... i represents the data point over the continent."

L.104 "leading to robust inversion" Comment: How is this a robust inversion? Do you edit the data (3-sigma)? I think you mean as you only have three parameters to fit? If so just remove robust and say you solve with OLS. Further, how was the gridding performed you need to elaborate on that.

• As suggested by the reviewer we have corrected and added a text to explain thoroughly how the data are gridded. It is worth noting that the results were interpolated on a map of 5km*5km grid only for visualization needs. The sentence now reads on line 105:

" The fit was done with the Ordinary Least Squares (OLS) method and all data points with time-series length less than 11 cycles (about a year) were discarded. The date at which $\sigma 0$ reaches a maximum within a seasonal cycle is obtained by converting the seasonal phase to fraction of a year (assuming a year counts for 360 days). We have found that along track analysis of the seasonal parameters of $\sigma 0$ showed no dependence to anisotropic effects. In the following, both ascending and descending measurements are mixed to keep a high density of observations and cover most AIS ( 1.9 million data points). For visualization needs, seasonal parameters are interpolated on a map of grid by averaging with Gaussian weights. We considered all data points within a 25 km radius and weighted with a decorrelation radius of 10 km."

L.108 "on snow" to "of snow"

• The correction has been done.

L.110 "echo and" remove echo

[Figure]

• The correction has been done.

L.111 "been previously. . ." to "been previously studied by Lacroix et al (2008)"

• The correction has been done.

L.112 Remove everything after Remy et al. (2015)

• The sentence has been modified as suggested. The sentence now reads on line 118 :

"The physics involved in both surface and volume echoes have been previously studied by Lacroix et al., (2008b)."

L.114 Rewrite first sentence to something "The snow surface can be modeled as. . ."

• The text has been rewritten. The sentence now reads on line 121:

"The snow surface can be modeled as a randomly rough surfaces because most naturally occuring surfaces are irregular."

L.115 "from rough" to "from a rough"

• The correction has been done.

L.116 Change to "The effective dielectric constant of the snow is"

• The correction has been done.

L.117 "of snow" to "of the snow" and "and ice" to "and the ice" and "prescribed" to "modelled". Remove "statistical geometries"

• The correction has been done. The sentence now reads on line 123 :

"The effective dielectric constant of the snow is a function of snow the density and the ice dielectric constant, while the roughness is usually modeled by two parameters: the surface correlation length (l) and the standard deviation of the surface elevation () (Ulaby et al., 1982)."

L.118 "height" Comment: Use either height or elevation

• We have changed "height to elevation" along the manuscript.

L.119 put "compared to the radar wavelength" into brackets, and add "," after "coefficients" and add "a" after "from".

• The correction has been done.

L.120 remove "the roughness has" and "as follows"

• The correction has been done. The sentence now reads on line 125:

" In the case of large standard deviations of the surface elevation () (compared to the radar wavelength), the backscattering coefficient, from a rough surface can be estimated assuming a Gaussian auto-correlation function (Ulaby et al., 1982):"

L.122 "at normal" to "at the normal" and add "angle" after "incident"

• The correction has been done.

L.125 Remove entire sentence "When the surface snow. . ." it's redundant.

• As suggested by the reviewer, the redundant sentence has been removed.

L.149 Remove "all"

• The correction has been done.

L.150 Remove "first"

• The correction has been done.

L.168 "appears in yellow" Comment: I think you should draw the boundary of the area in your figures to allow the reader to easier detect them.

• As suggested by the reviewer, we have revised all the figures and added the SP zones boundaries in the figures.

L.175 When using Julian days please also provide the months inside brackets

• The correction has been done.

L.195 please change "dhds" to "dh/ds" (s=sigma)

• The correction has been done.

L.203 Remove "the" before "snow" and put "it is poorly known" inside brackets

• The correction has been done.

L.211 "on volume" to "on the volume"

• The correction has been done.

L.212 "bands" to "band levels."

• The correction has been done.

L.217 "volume" to "medium"

• The correction has been done.

L.217 "Along increasing" Comment: Long sentence, should be re-written.

• The sentence has been rewritten. The sentence now reads on line 230:

"Also increasing snow grain size increases the scattering coefficient, which in turn increases the radar extinction in the medium. It results in a decrease of the radar wave penetration, therefore may limit the volume echo."

L.218 "which increases" to "which in turn increases"

• The correction has been done.

L.222 "temperature wave" maybe to "temperature gradient" and replace "to the subsurface of the" with "into". Further change "The volume echo variation" to "The variation in the volume echo"

• The correction has been done. The sentence now reads on line 233:

"This lag is related to the propagation of the temperature gradient from the surface into the snowpack. As the temperature controls the snow grain metamorphism and the radar wave penetration depth, the variation in the volume echo would be predominantly driven by the seasonal variations of snow temperature."

L.224 "echo increases" Comment: Increases in what; magnitude? Make clearer!

• The correction has been done. The sentence now reads on line 236:

"The magnitude of these echoes increase with increasing surface snow density, thus similar seasonal cycle of $\sigma 0$ would be expected at any frequency if snow density were the main driver."

L.239 remove "that of"

• The correction has been done.

L.244 "the increase" Comment: See L.224

• The correction has been done.

L.247 "which one among . . ." Sentence is worded strangely; please re-write

• As suggested , the sentence has been rewritten. The sentence now reads on line 262:

"However, in this study it is difficult to differentiate with certainty between the surface snow density and the snow surface roughness, that drives the seasonal cycle of the surface echo."

L.256 Remove "which" after matches, replace "greatest" with "large" and replace "of" before "radarsat" with "from".

• The correction has been done.

L.269 "the distribution" to "the spatial distribution"

• The correction has been done.

L.277 "in blowed" sounds strange please change sentence structure or remove.

• The correction has been done.

L.283 add months after Julian days and change "By blowing" into "Persistent winds" or similar

• The correction has been done. The sentence now reads on line 301:

"Cold and persistent winds may unusually accelerate the cooling of the surface snow temperature (Remy and Minster, 1991)."

L.288 "highest grain size vertical gradient" to "the highest vertical gradient in grain size"

• The correction has been done.

L.290 "difference observed" to "observed difference"

• The correction has been done.

L.294 This sentence sounds strange, maybe start something like this: "This study, using 35-day repeat radar altimetry data, allowed for. . ."

• The correction has been done. The sentence now reads on line 312:

"This study, using 35-day repeat radar altimetry data, allowed for carring out this spatial and temporal comparative analysis of the seasonal amplitude and date of maximum $\sigma 0$ at the S, Ku and Ka bands."

L.295 "used 8-year" to "used an 8-year"

• The correction has been done.

L.296 "band," to "band a" and "and 3-year" to "and a 3-year"

• The correction has been done.

L.297 "band all" to "band" and "covering 2002" to "covering the time period of"

• The correction has been done.

L.300 remove "on the AIS", add "with a" before "maximum" and "the" before "winter"

• The correction has been done.

L.302Remove "the" before "snow" and add "the seasonal changes in the" before "volume echo"

• The correction has been done.

L.303 Remove "the" before "snow properties"

• The correction has been done.

L.304 replace "because" with "due to" and "those properties" with "those parameters"

• The correction has been done.

L.306 Remove "which is between the S and Ka bands"

• The correction has been done.

L.307 "zones is" to "zones are"

• The correction has been done.

L.308 "or not" to "lack of"

• The correction has been done.